# A fast, scalable and versatile tool for analysis of single-cell omics data

Kai Zhang [1,6], Nathan R. Zemke[1,2], Ethan J. Armand[1,3] & Bing Ren [1,2,4,5] ✉

Single-cell omics technologies have revolutionized the study of gene regulation in complex tissues. A major computational challenge in analyzing these datasets is to project the large-scale and high-dimensional data into low-dimensional space while retaining the relative relationships between cells. This low dimension embedding is necessary to decompose cellular heterogeneity and reconstruct cell-type-specific gene regulatory programs. Traditional dimensionality reduction techniques, however, face challenges in computational efficiency and in comprehensively addressing cellular diversity across varied molecular modalities. Here we introduce a nonlinear dimensionality reduction algorithm, embodied in the Python package SnapATAC2, which not only achieves a more precise capture of single-cell omics data heterogeneities but also ensures efficient runtime and memory usage, scaling linearly with the number of cells. Our algorithm demonstrates exceptional performance, scalability and versatility across diverse single-cell omics datasets, including single-cell assay for transposase-accessible chromatin using sequencing, single-cell RNA sequencing, single-cell Hi-C and single-cell multi-omics datasets, underscoring its utility in advancing single-cell analysis.

Rapid advancements in single-cell omics technologies have enabled the analysis of the gene regulatory programs encoded in the genome at unprecedented resolution and scale[1]. Single-cell analysis of genomes, transcriptomes, open chromatin landscapes, histone modifications, transcription factor binding, DNA methylation, chromatin architecture, and so on, have provided valuable insights into the mechanisms governing cellular identity and regulation[1]. However, the extreme scale and complexity of single-cell omics data often present substantial computational challenges, necessitating the development of efficient, scalable and robust methods for data analysis[2].

A crucial step in analyzing single-cell omics data is to project the high-dimensional data into low-dimensional space while retaining the relative relationships between cells, a process known as dimensionality reduction. This step is key to the success of downstream analyses such as clustering, batch correction, data integration and visualization. Effective dimensionality reduction techniques are instrumental for visualization of distinct cell populations, identification of rare cell types and delineation of cell-type-specific transcriptional regulatory programs[2]. Currently, single-cell omics dimensionality reduction algorithms fall into two main categories: linear and nonlinear techniques. Linear dimensionality reduction algorithms, such as principal component analysis (PCA), used by SCANPY[3] and Seurat[4], for single-cell RNA-sequencing (scRNA-seq) data analysis, and latent semantic indexing (LSI) used by ArchR[5] and Signac[6] for single-cell assay for transposase-accessible chromatin using sequencing (scATAC-seq) data analysis, are popular due to their computational efficiency and scalability. However, these algorithms are not optimal for handling single-cell datasets with complex and nonlinear

[1]Department of Cellular and Molecular Medicine, University of California, San Diego School of Medicine, La Jolla, CA, USA. [2]Center for Epigenomics, University of California, San Diego School of Medicine, La Jolla, CA, USA. [3]Bioinformatics and Systems Biology Program, University of California, San Diego, La Jolla, CA, USA. [4]Ludwig Institute for Cancer Research, La Jolla, CA, USA. [5]Institute for Genomic Medicine, University of California, San Diego, La Jolla, CA, USA. [6]Present address: Westlake Laboratory of Life Sciences and Biomedicine, School of Life Sciences, Westlake University, Hangzhou, China. ✉e-mail: biren@health.ucsd.edu

structures, such as single-cell Hi-C (scHi-C) and single-cell multimodal omics datasets.

Nonlinear dimensionality reduction methods address these issues by more effectively capturing complex and often nonlinear cell relationships. Examples include latent Dirichlet allocation (LDA) used for scATAC-seq and scHi-C data[7,8], Laplacian-based algorithms used for scRNA-seq and scATAC-seq data[9–13], and various neural network models developed for scRNA-seq, scATAC-seq and scHi-C data[14–18]. Nonlinear dimensionality reduction methods have also become the standard approach for single-cell data visualization. For example, *t*-distributed stochastic neighbor embedding[19] and uniform manifold approximation and projection (UMAP)[20] are two widely used algorithms for this purpose, despite recent concerns regarding their reliability and validity[21]. While nonlinear methods excel in handling complex structures and projecting data into low-dimensional manifolds, they are generally computationally inefficient, with limited scalability. For instance, LDA relies on the Markov chain Monte Carlo algorithm for model training, which is slow to converge, computationally expensive and difficult to parallelize, making it difficult to be applied to large datasets[22]. Laplacian-based techniques like our previous work, SnapATAC[9], necessitate computing similarity matrices between all pairs of cells, which leads to quadratic memory usage increase with the number of cells[23,24]. Deep neural network models, known for their high training costs, often require specialized computational hardware such as graphics processing units (GPUs) to be computationally feasible.

In this study, we describe a nonlinear dimensionality reduction algorithm that achieves both computational efficiency and accuracy in discerning cellular composition of complex tissues from a broad spectrum of single-cell omics data types. The key innovation of our algorithm is the use of a matrix-free spectral embedding algorithm to project single-cell omics data into a low-dimensional space that preserves the intrinsic geometric properties of the underlying data. Unlike the conventional spectral embedding approach that requires the construction of the graph Laplacian matrix, a process that demands a storage space increasing quadratically with the number of cells, our algorithm achieves the same goal while avoiding this computationally expensive step. Specifically, we utilize the Lanczos algorithm[25] to derive eigenvectors while implicitly using the Laplacian matrix. This strategy substantially shortens the time and space complexity, making it linearly proportional to the number of cells in the single-cell data. To evaluate the accuracy and utility of our algorithm, we conducted extensive benchmarking using a variety of datasets that encompass diverse experimental protocols, species and tissue types. The results showed that our matrix-free spectral embedding algorithm outperforms existing methods in terms of speed, scalability and precision in resolving cell heterogeneity. Furthermore, we showed that our algorithm can be extended to diverse molecular modalities of single-cell omics datasets, revealing cell heterogeneity by leveraging complementary information from different single-cell omics data types.

We have implemented these algorithmic advancements in a Python package called SnapATAC2. This package is a major revamp of the original SnapATAC, offering substantial improvements such as increased speed, reduced memory usage, more reliable performance and a comprehensive analysis framework for diverse single-cell omics data. SnapATAC2 is freely available at https://github.com/kaizhang/SnapATAC2/.

## Results

### An overview of the SnapATAC2 workflow

SnapATAC2 is a comprehensive, high-performance solution for single-cell omics data analysis. Like the original SnapATAC[9], SnapATAC2 offers a wide range of functionalities to streamline the analysis of scATAC-seq data across multiple stages of the process. Moreover, SnapATAC2 is designed with flexibility in mind, intended for a variety of single-cell omics data types. For instance, its dimensionality reduction subroutine is readily applicable to scATAC-seq, scRNA-seq, single-cell DNA methylation and scHi-C data, showcasing its adaptability. To enhance performance and scalability, SnapATAC2 uses the Rust[26] programming language for executing computationally intensive subroutines and provides a Python[27] interface for seamless installation and user-friendly operation. This combination allows for efficient processing of large-scale single-cell omics data while maintaining accessibility for researchers across various levels of expertise. To further improve scalability when handling large-scale single-cell data, on-disk data structures and out-of-core algorithms are used whenever possible. These modifications facilitate the analysis of large datasets without overburdening system resources. Additionally, SnapATAC2 is modular and adaptable, and allows users to tailor their analysis to specific requirements and integrate with other software packages from the scverse[28] ecosystem, such as SCANPY[3] and scvi-tools[14].

The SnapATAC2 package is made up of four main parts: preprocessing, embedding/clustering, functional enrichment analysis and multimodal omics analysis (Fig. 1a). The preprocessing module handles raw BAM files, assesses data quality, creates count matrices and spots doublets, ensuring a strong base for downstream analysis. The core of SnapATAC2 is its embedding/clustering module, which introduces a new algorithm for reducing data dimensions. This module also helps in identifying unique cell clusters and revealing biological patterns. The functional enrichment module offers detailed data interpretation like differential accessibility and motif analysis. Finally, the multimodal omics analysis part allows for the examination of complex and multifaceted biological datasets, combining different types of biological data, and building networks to understand gene regulation.

### Efficient and accurate cell embedding for scATAC-seq data

Spectral embedding, also known as Laplacian eigenmaps, is a widely used technique for nonlinear dimensionality reduction[29]. This method boasts several key advantages, such as locality preservation, noise reduction and a natural connection to clustering[29]. Spectral embedding techniques leverage the spectrum (eigenvalues and eigenvectors) of the cell similarity matrix calculated from single-cell omics datasets to perform dimensionality reduction. However, the computation of this matrix is a rate-limiting step and a memory bottleneck, creating challenges for handling datasets consisting of large numbers of cells. For example, the memory usage of the similarity matrix for a dataset with one million cells is approximately 7 TB, far beyond the capacity of most computational servers. To address this barrier, we devised a matrix-free spectral embedding algorithm that efficiently computes eigenvectors using the Lanczos algorithm[25], eliminating the need for constructing a full similarity matrix (Fig. 1b and Methods). This method exhibits linear space and time usage relative to the input matrix size, resulting in a faster and memory-efficient approach for processing of large datasets. Notably, our algorithm avoids heuristic approximations, delivering precise solutions, distinguishing it from previous methods that generate approximate outcomes[10,11,30] (Methods).

To benchmark the performance of SnapATAC2, we generated synthetic scATAC-seq datasets with varying cell numbers and compared the scalability of the matrix-free spectral embedding algorithm to other widely used dimensionality reduction algorithms, such as LSI (used by ArchR[5] and Signac[6]), LDA (used by cisTopic[7]), PCA (used by EpiScanpy[31]) and classic spectral embedding with the Jaccard index (implemented in the original SnapATAC[9] package). In addition to these, we also considered deep neural network-based approaches, such as PeakVI[15], scBasset[16] and SCALE[17]. The benchmarks were conducted on a Linux server utilizing four cores of a 2.6 GHz Intel Xeon Platinum 8358 CPU. For neural network methods, we additionally used an A100 GPU to accelerate calculations and monitored the runtime over a total of 50 epochs, a commonly accepted minimum number of epochs required for algorithmic convergence. Our findings, illustrated in Fig. 1c, show that SnapATAC2, along with ArchR, Signac and EpiScanpy, had the least

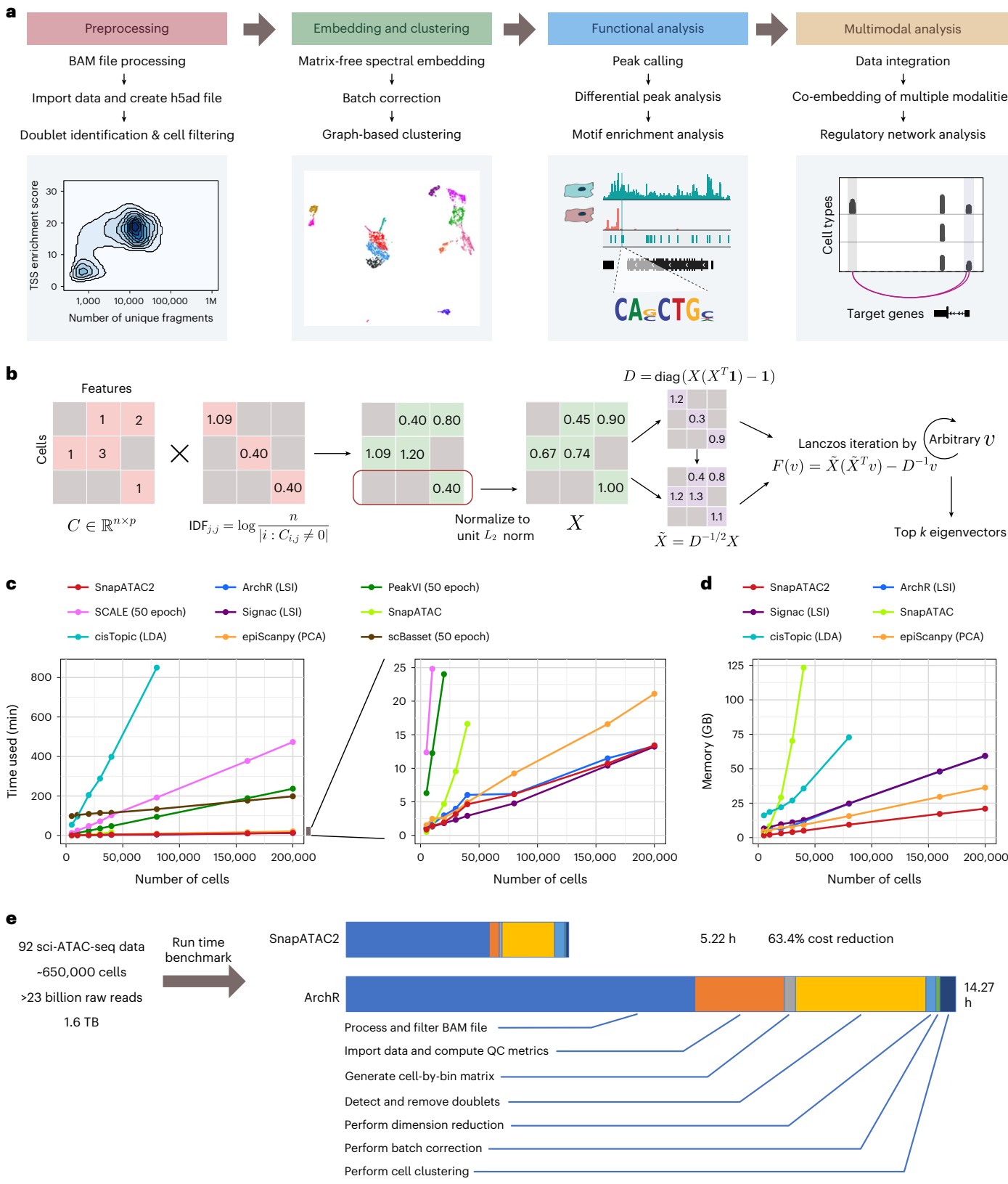

**Fig. 1 | SnapATAC2 enables comprehensive and scalable analysis of scATAC-seq data. a**, Overview of the SnapATAC2 Python package, featuring four primary modules: preprocessing, embedding/clustering, functional enrichment analysis and multimodal analysis. **b**, Schematic representation of the matrix-free spectral embedding algorithm in SnapATAC2, consisting of four main steps: feature scaling with inverse term frequency, row-wise $L_2$ norm normalization, normalization using the degree matrix and eigenvector calculation through the Lanczos algorithm[25]. **c**, Line plots comparing running times of various dimensionality reduction algorithms for scATAC-seq data. **d**, Line plots comparing memory usage of various dimensionality reduction algorithms for scATAC-seq data. Neural network-based methods were excluded from this comparison because their memory usage does not scale with the number of cells (Methods). **e**, Runtime comparison between ArchR and SnapATAC2 for end-to-end analysis of 92 raw BAM files produced by scATAC-seq experiments. TSS, transcription start site; QC, quality control.

increase in runtime as the number of cells in the dataset expanded. Neural network-based methods, despite their linear scalability, were considerably slower. For example, SnapATAC2 took only 13.4 min to analyze a dataset with 200,000 cells, whereas PeakVI needed approximately 4 h.

Regarding memory efficiency, SnapATAC2 stood out by requiring only 21 GB of memory to process 200,000 cells (Fig. 1d). In contrast, the original SnapATAC package showed limitations, encountering out-of-memory errors when handling over 80,000 cells on a server with 500 GB of available memory. cisTopic, although not constrained by memory, demonstrated the highest growth in runtime among all tested methods (Fig. 1c,d). We excluded neural network-based methods from memory usage comparisons, as their memory requirements do not scale with the cell count, thanks to the use of mini-batch training. Nevertheless, these methods do consume substantial memory proportional to the number of features (for example, peaks or genes). For instance, PeakVI, scBasset and SCALE exhausted the available memory on an A100 GPU with 40 GB when the feature count exceeded 500,000.

One of the aims of SnapATAC2 is to offer a wide-ranging analysis for scATAC-seq data, covering multiple stages of the process. ArchR has been previously cited as one of the most scalable and comprehensive software packages for similar tasks[32]. To evaluate how SnapATAC2 measures up against ArchR, we conducted side-by-side analyses across eight critical stages in scATAC-seq data processing. These include BAM file filtering and processing, data import, quality-control metric calculation, cell-by-bin matrix creation, doublet identification and removal, dimensionality reduction, batch correction and clustering. We utilized a human single-cell atlas of chromatin accessibility for this comparison[33]. This atlas, which we previously published, comprises 92 scATAC-seq samples, around 650,000 cells, and more than 23 billion raw reads, totaling a data size of 1.6 TB. According to our findings (Fig. 1e), SnapATAC2 completed the analysis in 5.22 h on a Linux server with eight CPU cores and 64 GB memory, while ArchR took 14.27 h for the same tasks. To summarize, at this data scale, SnapATAC2 is nearly three times faster than ArchR, leading to an approximate reduction in computational costs of 63.4%.

## SnapATAC2 is robust to noise and varying sequencing depths

We proceeded to assess the precision of our dimensionality reduction algorithm in identifying the relationships between cells, in comparison to other existing methods. For this purpose, we utilized a previously published benchmark dataset of synthetic scATAC-seq data[22], consisting of eight simulated datasets with varying sequencing depths (5,000, 2,500, 1,000, 500 and 250 reads per cell) and noise levels (0, 0.2 and 0.4). Each dataset contains 1,200 cells and includes the following six cell types: hematopoietic stem cells, common myeloid progenitors, erythroid cells, natural killer cells, CD4+ T cells and CD8+ T cells (Fig. 2a). After dimensionality reduction, we applied graph-based clustering using the Leiden algorithm[34] and assessed the clustering quality with the adjusted Rand index (ARI), which measures the similarity between two data clusterings and has been routinely used to assess the performance of clustering algorithms[22,35]. We hypothesized that high-quality embeddings should yield clusters consistent with the ground-truth cell-type labels, and hence resulted in high ARI scores.

Our findings, illustrated in Fig. 2b, reveal that SnapATAC2 consistently outperformed other methods across varying sequencing depths, achieving the highest ARI scores. For example, at a sequencing depth of 5,000 reads per cell, all tested algorithms accurately identified the six cell types, garnering ARI scores between 0.94 and 1.00. However, when the sequencing depth was reduced to 1,000 reads per cell, only SnapATAC2 and Signac maintained an ARI score above 0.9. Particularly, PeakVI was highly sensitive to sequencing depth, its ARI score plummeting to 0.006 at 250 reads per cell (Fig. 2b,c). In contrast, SnapATAC2 maintained a score of 0.47.

We observed similar robustness in performance when assessing noise levels. SnapATAC2 achieved perfect ARI scores (1.0) at all examined noise levels, followed closely by Signac and the original SnapATAC (Fig. 2d). In comparison, SCALE and PeakVI showed the most sensitivity to noise, with their ARI scores dropping to 0.57 and 0.46, respectively, at a noise level of 0.4 (Fig. 2d,e). Moreover, SnapATAC2 excelled in identifying rare cell populations in simulated datasets with variable cell-type abundances (Extended Data Fig. 1). In summary, our results demonstrate that SnapATAC2 is highly robust to both variable sequencing depths and noise levels, delivering consistently high-quality embeddings.

## Benchmarking SnapATAC2 with real scATAC-seq data

To rigorously evaluate SnapATAC2's performance in conditions that closely resemble real experimental data, we analyzed multiple publicly available scATAC-seq datasets[36–42]. These datasets span different technologies, species and tissue types (Table 1) and come with available cell-type labels. To ensure data reliability, we limited our analysis to datasets that have been broadly cited in the scientific literature.

We began our evaluation by comparing SnapATAC2 with other dimensionality reduction algorithms using a well-regarded human hematopoietic system scATAC-seq dataset[36]. This dataset is widely recognized as a benchmark for scATAC-seq analysis methods, including 2,034 hematopoietic cells profiled and subjected to fluorescence-activated cell sorting (FACS) from ten cell populations: hematopoietic stem cells, multipotent progenitors, lymphoid-primed multipotent progenitors, common myeloid progenitors, granulocyte–macrophage progenitors, granulocyte–macrophage progenitor-like cells, megakaryocyte–erythroid progenitors, common lymphoid progenitors, monocytes and plasmacytoid dendritic cells (Fig. 3a). To assess the bio-conservation quality of the cell embeddings generated by each method, we used a suite of metrics: the ARI, adjusted mutual information (AMI), cell-type average silhouette width (cell-type ASW)[43] and graph integration local inverse Simpson's index (graph iLISI)[43]. Detailed explanations of these metrics are available in the Methods. Our analysis revealed that SnapATAC2 outperformed the other eight methods examined, ranking highest based on the average scores across all four metrics (Fig. 3b,c). Notably, nonlinear methods such as cisTopic, PeakVI and scBasset followed SnapATAC2. This pattern was further substantiated across nine additional benchmark datasets (Fig. 3d and Extended Data Figs. 2–6), where nonlinear methods consistently outperformed their linear counterparts.

On average, SnapATAC2 achieved the top bio-conservation scores across all ten datasets and was followed by PeakVI, cisTopic and scBasset (Fig. 3d). Beyond excelling in cell-type identification, SnapATAC2 also presents several advantages over other high-performing methods like cisTopic and deep neural network-based algorithms. Specifically, SnapATAC2 can operate without the need for specialized hardware like GPUs, requires substantially less computational time, maintains robust performance across diverse datasets and eliminates the need for extensive hyperparameter tuning.

## SnapATAC2 is applicable to a wide range of omics data types

Spectral embedding is a versatile and effective technique across a broad spectrum of applications. We next explored whether this algorithm could be applied to other single-cell data types, such as scRNA-seq and scHi-C.

scHi-C data is notably sparse and exhibits an extraordinarily high dimensionality. Current computational methods struggle to fully utilize sparse scHi-C data for analyzing cell-to-cell variability in three-dimensional genome features. Therefore, we initially focused on scHi-C data and tested our method, SnapATAC2, on two datasets with multiple cell types or known cell-state information, including a sci-Hi-C dataset[8] made public by the 4D Nucleome Project (4DN) and a dataset from ref. 44. We converted scHi-C data into a cell-by-feature

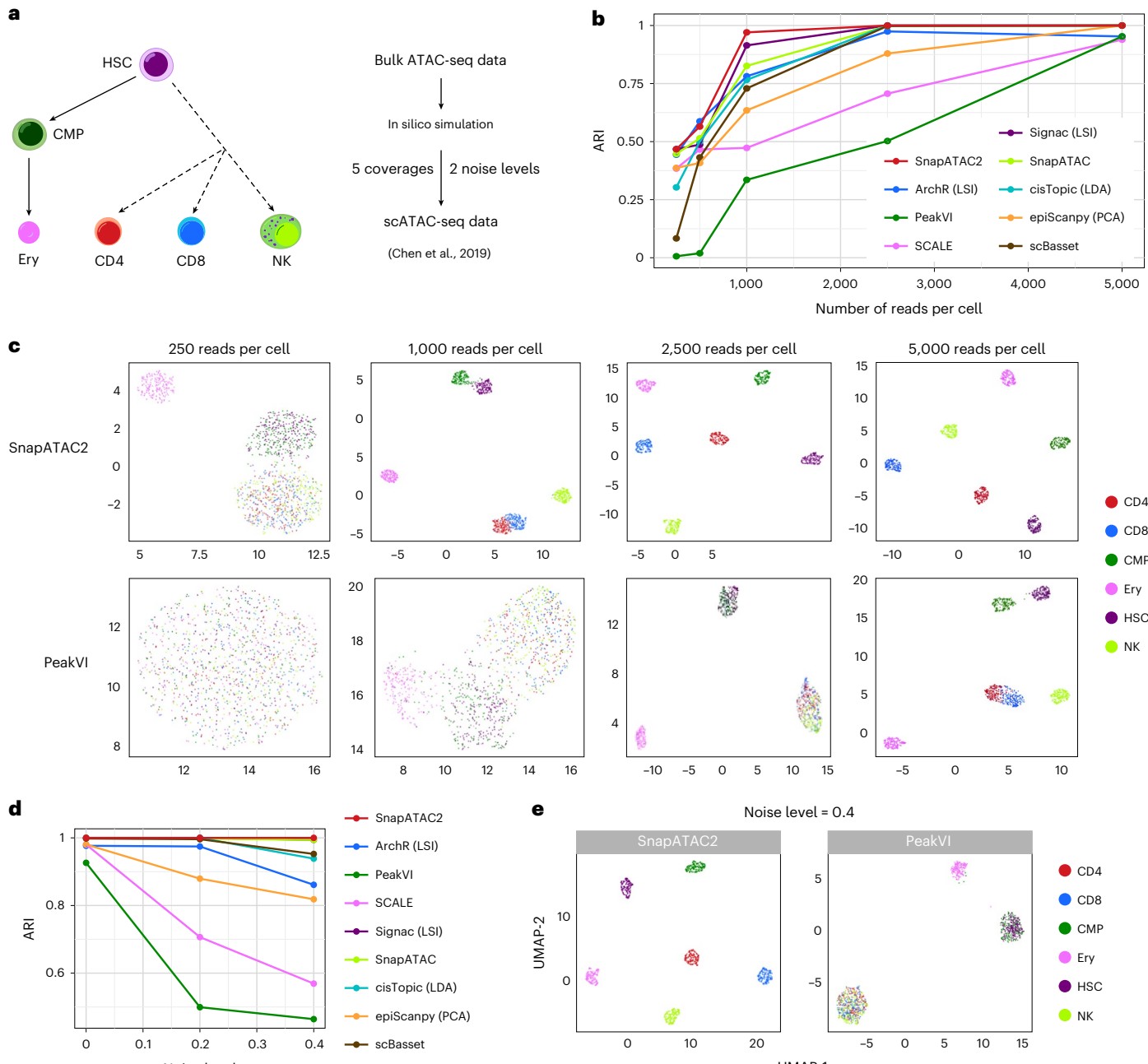

**Fig. 2 | SnapATAC2's dimensionality reduction algorithm is robust to various noise levels and sequencing depths. a**, Schema of the synthetic scATAC-seq datasets[22] used in the present study. **b**, Line plot showing the ARI (*y* axis) as a function of the number of reads per cell (*x* axis) for nine dimensionality reduction methods. **c**, UMAP visualization of the embeddings generated by the best performing method (SnapATAC2) and the worst performing method (PeakVI) for the simulated dataset with varying sequencing depths. Individual cells are color coded based on the cell-type labels indicated in **a**. **d**, Line plot showing the ARI (*y* axis) as a function of the noise level (*x* axis) for nine dimensionality reduction methods. **e**, UMAP visualization of the embeddings generated by the best performing method (SnapATAC2) and the worst performing method (PeakVI) for the simulated dataset at a noise level of 0.4. Individual cells are color coded based on the cell-type labels indicated in **a**. CMP, common myeloid progenitor; Ery, erythroid; HSC, hematopoietic stem cell; NK, natural killer.

count matrix by flattening the contact matrices of individual cells into vectors. This count matrix served as input for SnapATAC2's matrix-free spectral embedding algorithm. The resulting cell embeddings exhibited clear patterns corresponding to the underlying cell types and cellular states (Fig. 4a). We next compared the quality of cell embeddings generated by SnapATAC2 with three methods: Higashi[18], scHiCluster[45] and PCA. Our analysis revealed that SnapATAC2 achieved substantially higher bio-conservation scores than both scHiCluster and PCA on both datasets (Fig. 4a,b). Furthermore, it displayed performance on par with

Higashi (Fig. 4a,b), which is currently considered the state-of-the-art method in scHi-C analysis. What sets SnapATAC2 apart, especially when compared to Higashi, is its computational efficiency and accessibility. SnapATAC2 operates with a substantially reduced runtime and eliminates the need for specialized hardware. This makes it a highly practical choice for analyzing large-scale scHi-C datasets.

Extending our analysis, we applied SnapATAC2 to scRNA-seq datasets and compared its performance to two other methods commonly used for dimensionality reduction in this domain: scVI, a deep neural

**Table 1 | Curated scATAC-seq benchmark datasets used in the present study**

| Dataset | Protocol | Tissue | No. of cells | No. of cell types | No. of features | Reads per cell |
|---|---|---|---|---|---|---|
| Buenrostro et al.[36] | IFC scATAC-seq | Human bone marrow | 2,034 | 10 | 237,440 | 15,409 |
| 10x brain 5k[57] | 10x ATAC-seq | Mouse cortex | 2,317 | 10 | 155,093 | 38,282 |
| 10x PBMC 10k[57] | 10x Multiome | Human PBMCs | 9,631 | 19 | 107,194 | 20,479 |
| Chen et al.[38] | SNARE-seq | Mouse cerebral cortex | 9,190 | 22 | 241,757 | 2,641 |
| GSE194122 (ref. 41)[a] | 10x Multiome | Human PBMCs | 9,876 | 19 | 116,490 | 8,260 |
| Ma et al.[39] | SHARE-seq | Mouse skin | 32,231 | 22 | 340,341 | 4,152 |
| Trevino et al.[37] | 10x ATAC-seq | Human cerebral cortex | 8,981 | 13 | 467,315 | 16,519 |
| Yao et al.[40] | sci-ATAC-seq | Mouse primary motor cortex | 54,844 | 11 | 148,814 | 3,026 |
| Zemke et al., human[42a] | 10x Multiome | Human primary motor cortex | 15,284 | 20 | 380,517 | 16,854 |
| Zemke et al., mouse[42] | 10x Multiome | Mouse primary motor cortex | 45,089 | 19 | 330,448 | 28,880 |

[a]We used a subset of the original dataset due to profound batch effects (Methods).

network-based approach[14], and PCA, a standard linear method[3,4,35,46]. Across all five benchmark scRNA-seq datasets we tested[35], SnapATAC2 emerged as the top performer, generating cell embeddings that were most aligned with the underlying cell types (Fig. 4d,e and Extended Data Fig. 7). It outperformed both PCA and scVI, which ranked second and third, respectively. One distinct advantage of SnapATAC2 is its independence from data centering or scaling, steps that are typically essential for PCA-based analyses. We observed that PCA's performance suffered when applied to unscaled data, as evidenced by Fig. 4d. However, the process of scaling effectively converts a sparse matrix into a dense one, which can be both computationally expensive and limiting, especially for datasets with an extensive feature set. Overall, SnapATAC2's ability to function effectively without additional preprocessing steps like scaling not only maintains its computational efficiency but also makes it a more versatile and practical tool for high-dimensional data analysis.

SnapATAC2's dimensionality reduction algorithm is also applicable to single-cell DNA methylation data. When applied to 5-methylcytosine sequencing 2 (snmC-seq2) data generated in mouse pituitaries[47], SnapATAC2 produced cell embeddings that are largely consistent with the cell types identified by the original study (Extended Data Fig. 8). Notably, the method provided finer resolution for some cell types, such as somatotropes and lactotropes.

In conclusion, SnapATAC2 is a versatile and effective method for the analysis of various single-cell data types, including scATAC-seq, scHi-C, scRNA-seq and single-cell DNA methylation data. It demonstrates comparable or superior performance to existing methods, while offering practical advantages such as reduced runtime and no need for specialized hardware. Finally, we incorporated batch correction benchmarks into our evaluation, and SnapATAC2's performance remained robust and reliable (Extended Data Fig. 9 and Supplementary Tables 1 and 2). This further attests to its practicality in real-world scenarios where batch effects often pose a challenge.

## SnapATAC2 enables joint embedding of multi-omics data

The rapid expansion of single-cell multimodal omics technologies, such as 10x Multiome (ATAC/RNA-seq), Paired-Tag[48] and single-cell methyl-Hi-C/single-nucleus methyl-3C sequencing[44], has provided powerful tools for investigating gene regulatory mechanisms. We therefore investigated the applicability of our algorithm to single-cell multimodal omics data. Multi-view spectral embedding is an extension of spectral embedding, which enables the joint embedding of multiple data representation views. This method has demonstrated its ability to harness complementary information from individual views and enhance performance in downstream analyses, making it an ideal candidate for analyzing single-cell multi-omics data. The multi-view spectral embedding process typically consists of three steps: first, a similarity or

kernel matrix is calculated from each view; second, a joint kernel matrix is constructed by combining or co-regularizing the kernel matrices in a certain manner; and lastly, spectral embedding is performed using the joint kernel matrix. In this study, we opted for kernel addition to combine the kernel matrices, as it has shown to be an effective method for achieving excellent clustering results[48,49]. Moreover, kernel addition enables the extension of the matrix-free spectral embedding algorithm to multi-view spectral embedding while maintaining the linear time and space complexity of the algorithm (Methods).

We applied this matrix-free multi-view spectral embedding algorithm to a 10x Genomics Multiome dataset, which jointly profiles chromatin accessibility and the transcriptome for 9,181 human peripheral blood mononuclear cells (PBMCs). To better evaluate the algorithm's performance, we first annotated the cells according to a previously published single-cell atlas of human PBMCs[4]. To compare the performance of joint embedding with individual views, we also performed spectral embedding on each modality separately. Our findings reveal that, while independent unsupervised analyses of RNA and ATAC data generated predominantly consistent cell classifications, there were notable differences (Fig. 5a,b). For instance, CD8$^+$ and CD4$^+$ T cells were close to each other when analyzing the transcriptome but separated clearly in the ATAC data (Fig. 5a,b). Conversely, intermediate and memory B cells partially overlapped when analyzing the ATAC data but were more distinguishable in the transcriptomic data (Fig. 5a,b). In comparison to the separate analysis of either modality, multi-view spectral embedding using both modalities clearly separated CD4$^+$ and CD8$^+$ T cells and uncovered subtle heterogeneity within B cells. Overall, the joint embedding of ATAC and RNA data enhanced the separation of cell types and revealed subtle heterogeneity within cell types, as evidenced by the increased silhouette scores across different cell types (Fig. 5b).

In our pursuit to further benchmark the performance of SnapATAC2, we compared it against other joint embedding techniques, specifically MIRA[50], Cobolt[51] and MOFA+[52]. Using the same 10x Genomics Multiome dataset, SnapATAC2 consistently ranked highest in bio-conservation scores across all four evaluation metrics (Fig. 5c). To broaden the scope of our comparative analysis, we also incorporated a dataset profiling trimethylated histone H3 Lys 27 (H3K27me3) occupancy and gene expression in 10,180 cells from the mouse frontal cortex[53]. Once again, SnapATAC2 emerged as the top-performing method, achieving the highest average bio-conservation score (Fig. 5d). Beyond its exceptional accuracy, SnapATAC2 also showed unparalleled scalability. Across both datasets, it drastically outperformed MIRA, Cobolt and MOFA+ in computational speed and memory efficiency, running more than 30 times faster than the next best method (Fig. 5c,d). In summary, these results validate SnapATAC2's excellent performance not only in bio-conservation quality but also in computational efficiency,

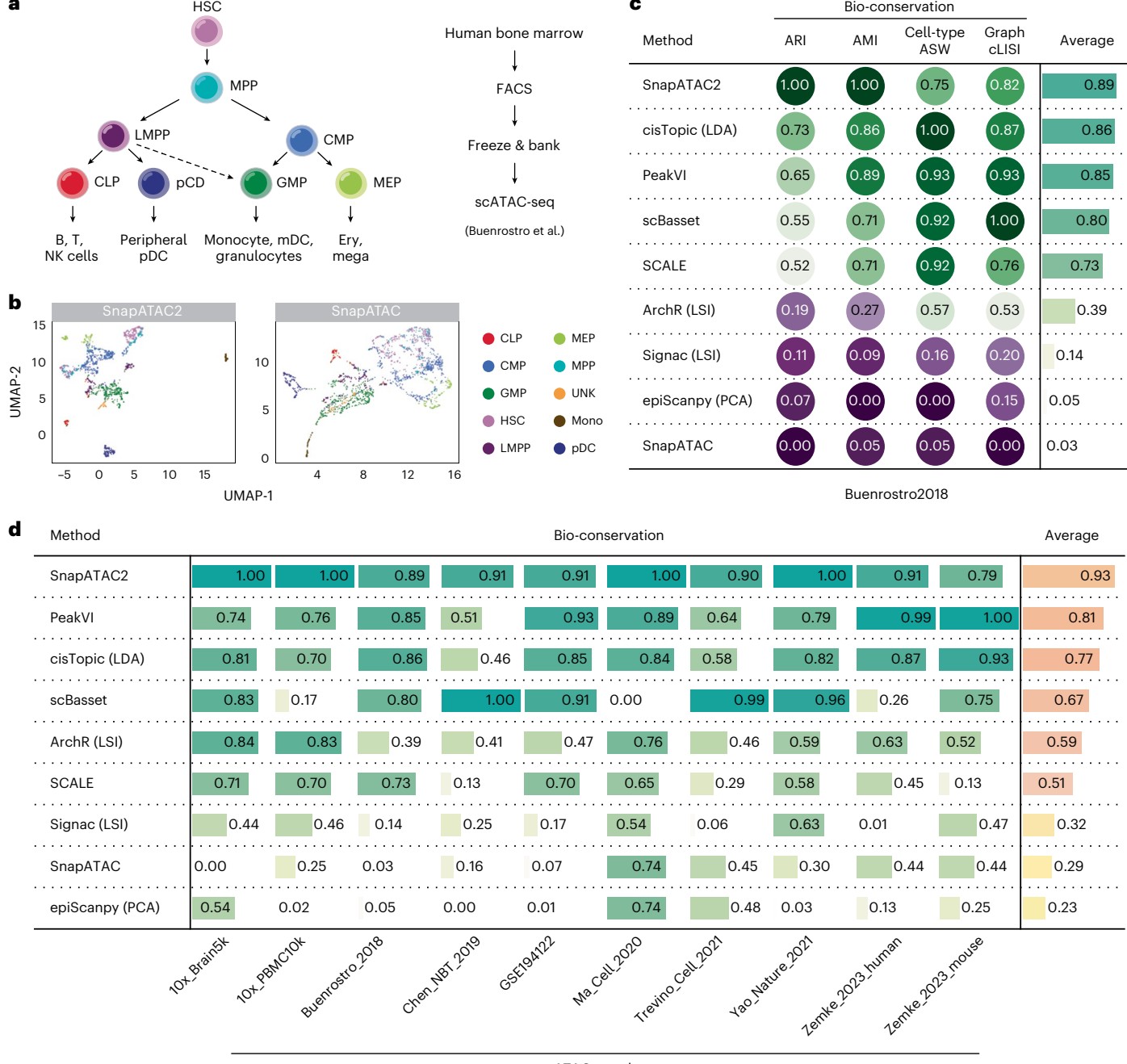

**Fig. 3 | Benchmarking of SnapATAT2 and other dimensionality reduction algorithms using real scATAC-seq data with cell labels. a**, Overview of cell types analyzed in the Buenrostro et al. scATAC-seq dataset. **b**, UMAP visualization of the embeddings generated by the best performing method (SnapATAC2) and the worst performing method (original SnapATAC) for the Buenrostro et al. dataset. Individual cells are color coded based on the cell-type labels indicated in **a**. **c**, Table displaying normalized scores (0–1 range) of four metrics used to evaluate each method's bio-conservation on the Buenrostro et al. dataset.

A score of 1 indicates optimal performance. See Methods for metric details. **d**, Table displaying the bio-conservation scores of nine dimensionality reduction methods across ten benchmark datasets (Extended Data Figs. 2–6). CLP, common lymphoid progenitor; GMP, granulocyte–macrophage progenitor; LMPP, lymphoid-primed multipotent progenitor; MEP, megakaryocyte–erythroid progenitor; mono, monocyte; MPP, multipotent progenitor; pDC, plasmacytoid dendritic cell.

making it a highly robust and scalable solution for analyzing complex single-cell multi-omics data.

## Discussion

In the present study, we describe SnapATAC2 for the analysis of a diverse array of single-cell omics data. The performance of SnapATAC2 exceeds that of existing dimensionality reduction methods in terms of accuracy,

noise robustness and scalability, thus providing researchers with a powerful tool for investigating gene regulatory programs using single-cell genomics, transcriptomics and epigenomics analysis.

SnapATAC2 offers a unique advantage in its seamless compatibility with other software tools widely used in the single-cell analytics ecosystem. By adopting the AnnData format, it facilitates effortless integration with established packages like SCANPY, scvi-tools and

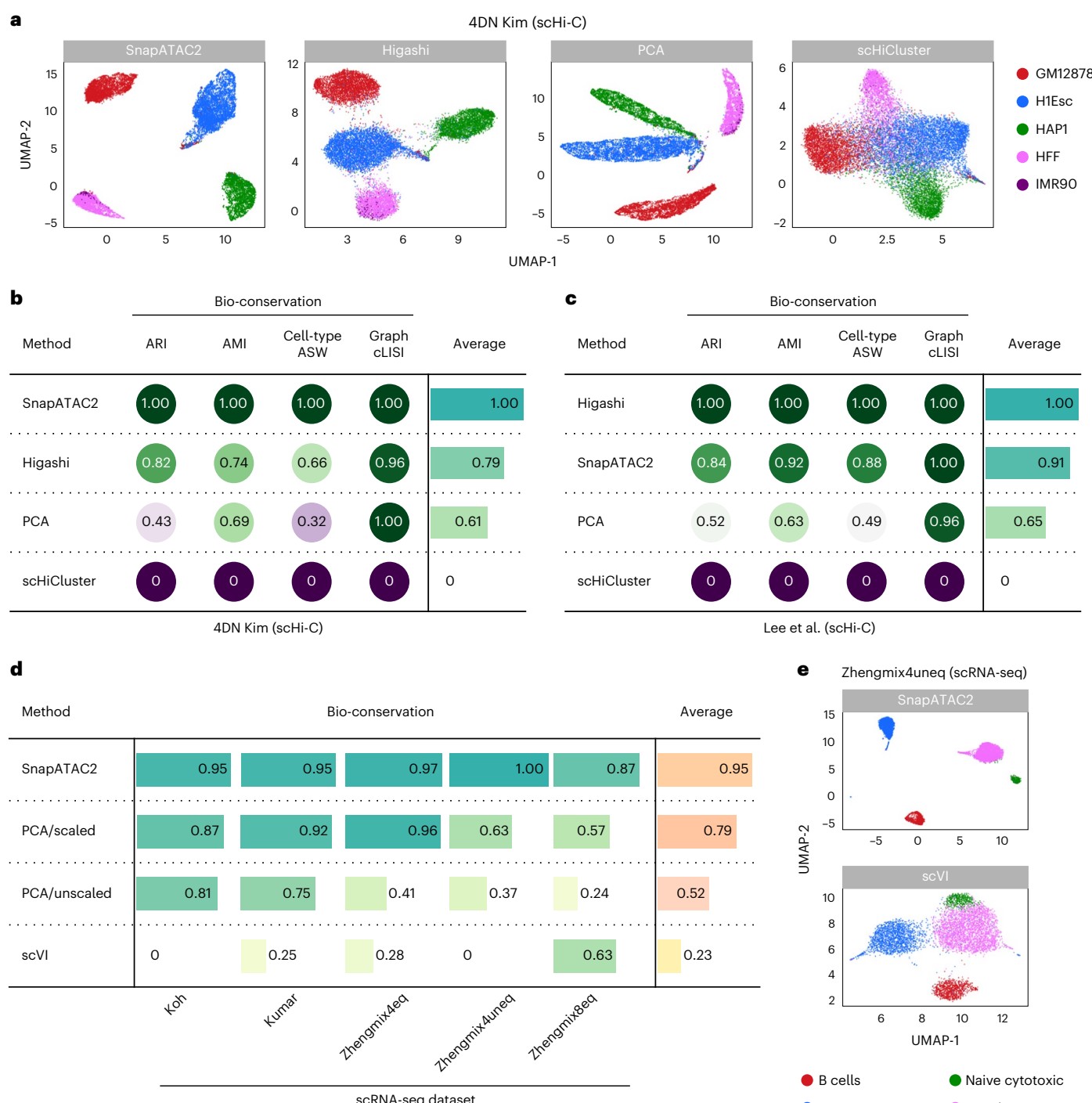

**Fig. 4 | SnapATAC2 demonstrates superior performance over other methods on scHi-C and scRNA-seq datasets. a**, UMAP visualization of the embeddings generated by Higashi, SnapATAC2, scHiCluster and PCA for the 4DN dataset by Kim et al. Cells are color coded based on cell-type labels. **b**, Table displaying normalized scores (0–1 range) of four metrics used to evaluate each method's bio-conservation on the 4DN dataset by Kim et al.[8]. **c**, Table displaying normalized scores (0–1 range) of four metrics used to evaluate each method's bio-conservation on the Lee et al. dataset. **d**, Table displaying the bio-conservation scores of four dimensionality reduction methods across five benchmark datasets (Extended Data Fig. 7). **e**, UMAP visualization of the embeddings produced by the best performing method (SnapATAC2) and the worst performing method (scVI) for the Zhengmix4uneq dataset[35]. Cells are color coded according to cell-type labels.

SCENIC+[54]. This feature is especially advantageous for researchers seeking to carry out specialized analyses, such as data imputation or trajectory inference, thereby enhancing the core functions of SnapATAC2.

The key innovation of SnapATAC2 lies in its matrix-free spectral embedding algorithm for dimensionality reduction. While numerous algorithms have been proposed to expedite spectral embedding[10,11,23,30], our algorithm stands out as it does not rely on sub-sampling or approximations, delivering the exact solution. This algorithm not only outperforms current methods in identifying cell clusters and heterogeneity but also maintains computational efficiency, making it highly suitable for large-scale single-cell omics data analysis. Furthermore,

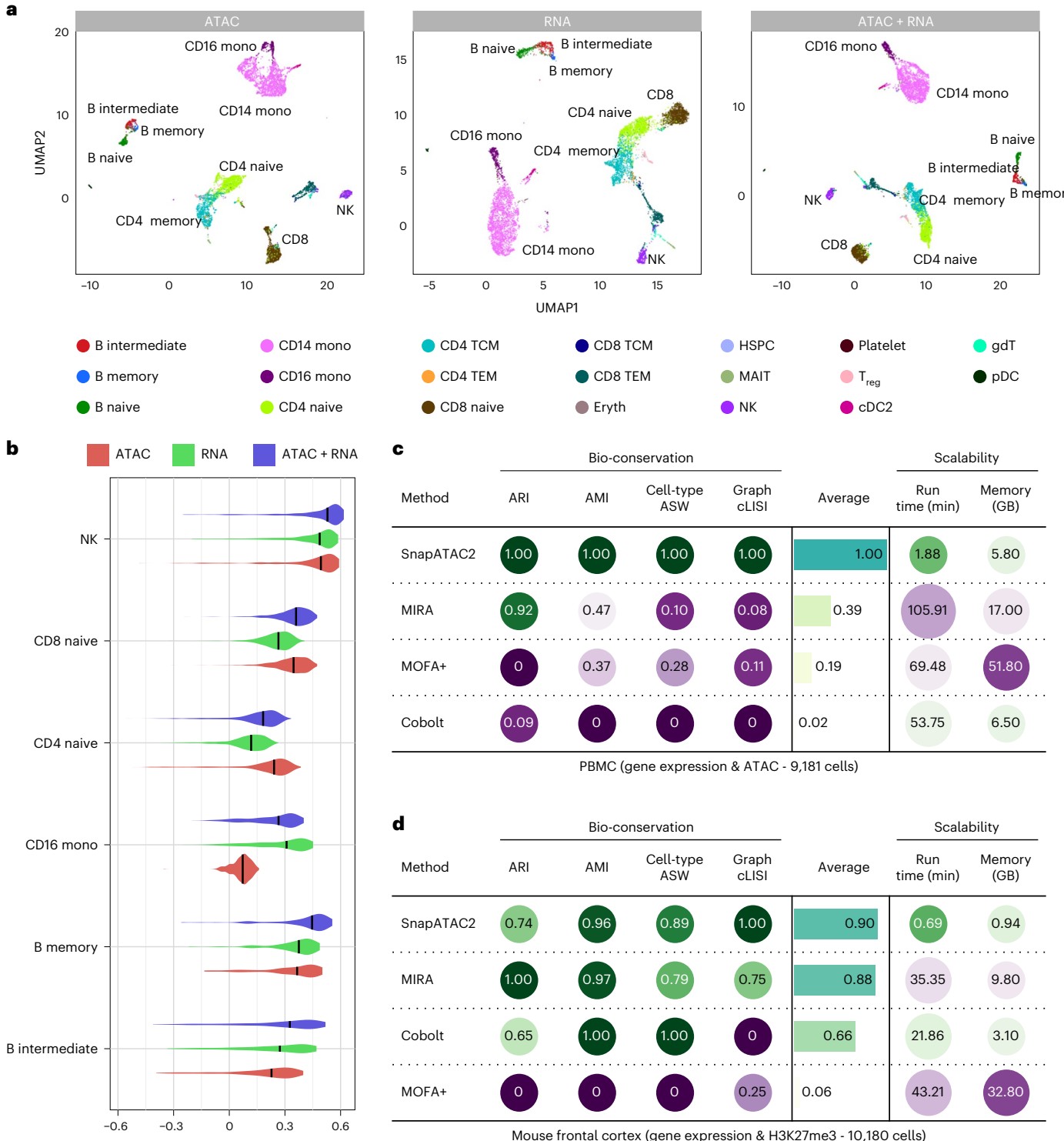

**Fig. 5 | SnapATAC2 enables robust joint embedding of single-cell multi-omics data. a**, UMAP visualization of the embeddings generated by SnapATAC2 using ATAC modality (left), RNA modality (middle) or both modalities (right) on a 10x Genomics Multiome dataset consisting of 9,181 human PBMCs. Cells are color coded based on cell-type labels. **b**, Violin plot comparing the silhouette scores of selected cell types derived from embeddings produced by the ATAC modality, the RNA modality or both modalities. The black line within each curve indicates the median value. **c**, Table comparing bio-conservation and scalability metrics of various joint embedding methods on 10x Genomics Multiome data from human PBMCs. **d**, Table comparing bio-conservation and scalability metrics of various joint embedding methods on Paired-Tag data from mouse frontal cortex.

we demonstrated the versatility of the matrix-free spectral embedding algorithm by applying it to various single-cell data types, including scATAC-seq, scRNA-seq, single-cell DNA methylation, scHi-C and single-cell multi-omics data.

One limitation of the matrix-free spectral embedding algorithm is that it currently is implemented using only cosine function-based similarity. For some data types, researchers may prefer to use other metrics to quantify the cell-to-cell similarity. For instance, in our findings, the

Euclidean distance yielded more accurate results for the protein expression data used in cellular indexing of transcriptomes and epitopes by sequencing experiments[55]. Future developments could extend the matrix-free algorithm to accommodate other similarity metrics. For instance, a potential solution involves leveraging a small set of landmark points to transform the given data into sparse feature vectors[56], followed by the application of the scalable matrix-free spectral embedding algorithm. In conclusion, SnapATAC2 represents a substantial advancement in single-cell data analysis, offering an accessible, scalable and high-performance solution for researchers studying epigenomics. With continued development and optimization, SnapATAC2 has the potential to become a general tool in single-cell multi-omics data analysis, ultimately facilitating new biological discoveries.

## Online content

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

## Methods

### Dimensionality reduction using spectral embedding

In this section, we outline the core algorithms used to perform dimensionality reduction in the SnapATAC2 package. We first describe the preprocessing steps and then the classic spectral embedding method that works for arbitrary similarity metrics. Finally, we describe the matrix-free spectral embedding algorithm that works only for cosine similarity, but substantially decreases the running time and memory usage. Note the steps described below can be accomplished using the 'snapatac2.tl.spectral' function from the SnapATAC2 package.

**Preprocessing.** Given a cell-by-feature count matrix $C \in \mathbb{R}^{n \times p}$, we first scale the columns of the matrix by the inverse document frequency (IDF). The IDF of a column or a feature $f$ is defined by $idf(f) = \log \frac{n}{1 + |i: C_{i,f} \neq 0|}$.

**Spectral embedding.** Assuming the cell-by-feature count matrix $C$ has been preprocessed according to the procedures described above, in classic spectral embedding, we first compute the $n \times n$ pairwise similarity matrix $W$ such that $W_{ij} = \delta(C_{i*}, C_{j*})$, where $\delta : \mathbb{R}^p \times \mathbb{R}^p \to \mathbb{R}$ is the function defining the similarity between any two cells. Typical choices of $\delta$ include the Jaccard index and the cosine similarity. We then compute the symmetric normalized graph Laplacian $L_{sym} = I - D^{-1/2} W D^{-1/2}$, where $I$ is the identity matrix and $D = \text{diag}(W\mathbf{1})$. The bottom eigenvectors of $L_{sym}$ are selected as the lower-dimensional embedding. The corresponding eigenvectors can be computed alternatively as the top eigenvectors of the similarly normalized weight matrix: $\tilde{W} = D^{-1/2} W D^{-1/2}$.

**Matrix-free spectral embedding with cosine similarity.** In this section, we introduce a matrix-free algorithm for spectral embedding that avoids calculating the similarity matrix. This approach is specifically designed for cosine similarity. The cosine similarity between two vectors $A$ and $B$ is given by $S_c(A, B) = \frac{A \cdot B}{\|A\| \|B\|}$. To express the cosine similarity using matrix operations, we first rescale the nonnegative count matrix $C$ to obtain a new matrix $X$, such that the rows of $X$ have unit $L_2$ norm. Consequently, the cosine similarity matrix between rows of $X$ can be represented as $XX^T$.

In traditional spectral clustering algorithms, it is necessary to set the diagonals of the similarity matrix to zero[58]. This can be accomplished by subtracting the identity matrix from the similarity matrix, resulting in the final similarity matrix $W = XX^T - I$. The degree matrix can then be calculated as $D = \text{diag}((XX^T - I)\mathbf{1}) = \text{diag}(X(X^T\mathbf{1}) - \mathbf{1})$. The normalized similarity matrix, denoted as $\tilde{W}$, can then be computed as follows:

$$\tilde{W} = D^{-1/2} XX^T D^{-1/2} - D^{-1} = \tilde{X}\tilde{X}^T - D^{-1}$$

where $\tilde{X} = D^{-1/2}X$. It is important to note that $\tilde{X}$ has the same dimensions as $X$, and if $X$ is sparse, $\tilde{X}$ preserves the sparsity pattern of $X$. Conventional spectral embedding algorithms compute $\tilde{W}$ and select its top eigenvectors as the lower-dimensional embedding. Previous work has attempted to compute the top eigenvectors of an approximation of $\tilde{W}$ to avoid the need for computing the full similarity matrix[30]. In other studies[10,11], the authors chose not to set the diagonals of the similarity matrix to zero. Consequently, the eigendecomposition of $\tilde{W}$ is equivalent to the singular value decomposition (SVD) of $\tilde{X}$, which can be computed efficiently. However, our benchmarking reveals that setting the diagonal of $W$ to zero is necessary as it substantially improves the embedding quality.

Unlike previous work, we offer an exact solution to the problem. We apply the Lanczos algorithm[25], an iterative method for computing the top eigenvectors of a symmetric matrix, to our problem without ever calculating $\tilde{W}$. This requires computing the matrix–vector product between $\tilde{W}$ and $\mathbf{v}$ in each iteration, as follows: $\tilde{W}\mathbf{v} = \tilde{X}(\tilde{X}^T\mathbf{v}) - D^{-1}\mathbf{v}$, where $\mathbf{v}$ is the current solution to the eigenvalue problem and is

iteratively refined by the Lanczos algorithm. By using the specific order of operations shown in the formula, we can reduce the computational cost of the matrix–vector product to $2z + n$, where $n$ is the number of rows in $X$ and $z$ is the number of nonzero elements in $X$. In comparison, performing this operation on the full similarity matrix requires $n^2$ computations, which is prohibitively expensive for a large number of cells. Thus, our matrix-free method is substantially faster and more memory efficient. The pseudocode for our algorithm is shown in Extended Data Fig. 10a.

### Nyström method for out-of-sample embedding

The matrix-free method described above is very fast and memory efficient. However, for massive datasets with hundreds of millions of cells, storing the cell-by-feature count matrix itself may already be a challenge. To circumvent this memory constraint, we choose to sample a subset of cells from the full dataset and use these as landmarks to perform out-of-sample embedding using the Nyström method[24,59]. The pseudocode for this algorithm and detailed benchmark comparisons can be found in Extended Data Fig. 10b,c, Supplementary Fig. 1 and Supplementary Note 1.

### Multi-view spectral embedding

In this section, we extend our matrix-free spectral embedding method to perform dimensionality reduction on multimodal single-cell data. Assume we have data in multiple views, for example, chromatin accessibility and gene expressions, represented by a sequence of count matrices $\{X_i, X_2, \ldots, X_k\} \in \mathbb{R}^{n \times p_k}$. Our objective is to obtain a low-dimensional representation of the data while preserving cell similarity in each view using the spectral embedding method. One approach involves calculating the similarity matrix for each view, normalizing them and subsequently summing them. The resulting matrix is then used to compute the spectral embedding. This straightforward strategy has been effective in revealing clusters in prior research[48,49]. However, it necessitates the computation of the similarity matrix for each view, which is computationally demanding. Here, we present an algorithm that is efficient in both time and space for computing this embedding.

We first normalize each $X_i$ such that the rows of $X_i$ have unit $L_2$ norm. We then define $X$ as the horizontally concatenated view of the sequence of matrices,

$$X = \left( \sqrt{\frac{\lambda_1}{\|W_1\|_F}} X_1 \quad \sqrt{\frac{\lambda_2}{\|W_2\|_F}} X_2 \quad \cdots \quad \sqrt{\frac{\lambda_k}{\|W_k\|_F}} X_k \right)$$

where $\lambda_k$ is the user-defined weights measuring the relative importance of each view; $W_k = X_k X_k^T - I$ is the similarity matrix of the $k$-th view; $\|W_k\|_F$ is the Frobenius norm of $W_k$. We can see that,

$$XX^T - \sum_k \frac{\lambda_k}{\|W_k\|_F} I = \sum_k \frac{\lambda_k}{\|W_k\|_F}(X_k X_k^T - I)$$

$$= \sum_k \lambda_k \frac{W_k}{\|W_k\|_F}$$

Without loss of generality, we can assume $\sum_k \frac{\lambda_k}{\|W_k\|_F} = 1$. In practice, this can be achieved by normalizing $\lambda_k$. The above equation can now be written as,

$$XX^T - I = \sum_k \lambda_k \frac{W_k}{\|W_k\|_F}$$

Therefore, the matrix $W = XX^T - I$ is a linear combination of the normalized similarity matrices of the individual views. To compute the spectral embedding of $W$, it suffices to apply the matrix-free spectral embedding method described above to the concatenated view $X$. This algorithm is implemented in the 'snapatac2.tl.multi_spectral' function from the SnapATAC2 package. The pseudocode for this algorithm is shown in Extended Data Fig. 10d.

## Eigenvector selection in spectral embedding

Not all eigenvectors produced by spectral embedding are informative and relevant for clustering tasks. Selecting appropriate eigenvectors is essential, as using uninformative or irrelevant ones can lead to suboptimal clustering results. We found that the widely used elbow method for determining the number of eigenvectors is not consistently reliable in practice. To identify relevant eigenvectors, we propose a simple heuristic based on the eigenvalues of the graph Laplacian matrix. In this approach, each eigenvector is weighted by the square root of its corresponding eigenvalue, and these weighted eigenvectors are then used for further analyses.

## Overview of the benchmarking process

In this study, we conducted a thorough evaluation of SnapATAC2, focusing on its dimensionality reduction capabilities across a range of datasets, spanning scATAC-seq, scHi-C, scRNA-seq and multiome data. Moreover, we scrutinized the performance of SnapATAC2's batch effect correction features. The subsequent sections offer an in-depth overview of the datasets utilized, the benchmarking procedures used and the metrics applied for this comprehensive assessment.

## Preparing scATAC-seq benchmarking datasets

**Simulated scATAC-seq datasets.** We obtained eight simulated scATAC-seq datasets from a prior study[22], presented as cell-by-peak matrices. These datasets were derived from well-annotated bulk ATAC-seq datasets from bone marrow, with variations in noise levels and read coverages. Specifically, a noise parameter, ranging from 0 to 1, represented the fraction of reads appearing in a random peak from a sorted population, which was then used to produce the peak-by-cell matrices. The remaining reads were allocated based on the bulk sample's distribution. A matrix with a noise level of 0 perfectly retained the cell-type specificity of the reads within peaks, while a matrix with a noise level of 1 lacked any distinguishing information about cell types based on the reads within peaks. The simulated datasets featured three noise levels: none (0), moderate (0.2) and high (0.4). The clean dataset (zero noise level) also spanned five read coverages per cell: 5,000, 2,500, 1,000, 500 and 250 fragments. The datasets utilized predefined peak regions sourced from bulk ATAC-seq data.

**Curated scATAC-seq datasets.** For further benchmarking analysis, we curated ten additional scATAC-seq datasets (Table 1). For each dataset, we assembled a cell-by-peak count matrix using the annotated cells and peaks specified in the respective publications. We also sourced cell labels from these publications. In preprocessing all the datasets, we eliminated peaks that were absent in all cells. While we generally retained all cells from the datasets, exceptions were made for the GSE194122 and Zemke_human datasets. In these cases, the data were generated using multiple donors or protocols, leading to pronounced batch effects. To ensure that our evaluation was not skewed by these batch effects, we opted to use only a subset of cells from these two datasets, specifically those originating from a consistent donor or protocol. It is worth noting that the full versions of these two datasets were used in evaluating batch effect correction methodologies.

## Comparing dimensionality reduction methods on scATAC-seq data

We utilized the cell-by-peak count matrices from the aforementioned benchmarking datasets to assess various dimensionality reduction techniques. Unlike other tasks, we did not implement feature/peak selection or scaling for the ATAC tasks, as these steps are not customary in an ATAC workflow[43]. We executed the nine selected dimensionality reduction methods using their default settings, as specified in relevant tutorials or their associated research methodologies. The resulting lower-dimensional cell embeddings were then assessed using four distinct metrics, further elaborated upon in 'Benchmarking metrics'.

Typically, we fixed the dimensionality at 30, as it effectively captures most of the data variance. For methods that require fine-tuning of dimensionality or component count, like cisTopic, we followed the recommendations provided in their respective publications to ascertain the optimal dimensionality. A comprehensive elucidation on the operational specifics of each method is provided below.

**ArchR.** ArchR (version 1.0.1) is an R package for analyzing scATAC-seq data. To generate the lower-dimensional embedding of the data, we used the 'ArchR:::.computeLSI' function with the default parameters. The output dimension was set to 30. After performing the SVD, ArchR scales the singular vectors by the singular values. As a result, component selection is not necessary, so we used all 30 dimensions for downstream analysis. Note that ArchR includes three variants of the LSI algorithm: 'TF-logIDF', 'log(TF-IDF)' and 'logTF-logIDF'. Although we have benchmarked all three variants, we only report the results for the 'log(TF-IDF)' variant in the main text as it is the default setting.

**Signac.** Signac (version 1.6) is an R package for analyzing scATAC-seq data. To generate the lower-dimensional embedding of the data, we used the 'Signac:::RunTFIDF.default' and 'Signac:::RunSVD.default' functions with the default parameters. The initial output dimension was set to 30 and we used the elbow method to select the number of components retained for downstream analysis. Note Signac includes four variants of the LSI algorithm: 'IDF', 'TF-logIDF', 'log(TF-IDF)' and 'logTF-logIDF'. Although we have benchmarked all four variants, we only report the results for the 'log(TF-IDF)' variant in the main text as it is the default setting.

**EpiScanpy.** EpiScanpy (version 0.4.0) is a Python package for analyzing scATAC-seq data. We first normalized the count matrix using 'episcanpy.pp.normalize_per_cell' and 'episcanpy.pp.log1p' functions with the default parameters. We then used the 'episcanpy.pp.pca' function to generate the lower-dimensional embedding of the data. The initial output dimension was set to 30 and we used the elbow method to select the number of components retained for downstream analysis.

**SCALE.** SCALE (version 1.1.2) is a Python package for performing dimensionality reduction on scATAC-seq data. We used the command 'SCALE.py' with following parameters to generate the lower-dimensional embedding: '--min_peaks 0 --min_cells 0 -i 30'. Additionally, as we knew the number of cell types in the benchmarking datasets, we set the '-k' parameter (the number of clusters) to the true number of cell types.

**PeakVI.** PeakVI (version 0.19.0) is a Python package for performing dimensionality reduction on scATAC-seq data. We used the 'scvi.model.PEAKVI' function to create a model with the default parameters. The dimensionality of the latent variable was set to 30.

**scBasset.** scBasset (GitHub: c15bec3a73fa1e04822db723338d-234ca9d384ce) is a Python package for performing dimensionality reduction on scATAC-seq data. We followed the instructions in the scBasset GitHub repository to generate the lower-dimensional embedding of the data. The dimensionality of the latent variable was set to 30.

**pycisTopic.** pycisTopic (GitHub: 242c2a47aad475250f8ab-b2469a0e36085d6e460) is a Python package for analyzing scATAC-seq data. To generate the lower-dimensional embedding of the data, we first created a model using the 'create_cistopic_object' function with the default parameters. We then used 'run_cgs_models' to train the model with the following parameters: 'n_iter = 300, alpha = 50, alpha_by_topic = true, eta = 0.1, eta_by_topic = false'. We trained six models with different dimensions of the latent variable: 5, 10, 15, 20, 25 and 30. We then used the 'evaluate_models' function to select the best model for downstream analysis.

**SnapATAC.** SnapATAC (version 1.0) is an R package for analyzing scATAC-seq data. For datasets with less than 20,000 cells, we used the 'SnapATAC::runDiffusionMaps' function with the default parameters to generate the lower-dimensional embedding of the data. For datasets with more than 20,000 cells, running 'SnapATAC::runDiffusionMaps' on the full dataset requires a large amount of memory. In this case, we applied 'SnapATAC::runDiffusionMaps' on a subset of the data and then used the 'SnapATAC::runDiffusionMapsExtension' function to generate the lower-dimensional embedding of the full dataset. The output dimension was set to 30. We used the 'SnapATAC:::weightDimReduct' to scale eigenvectors by their corresponding eigenvalues. The scaled eigenvectors were then used for downstream analysis.

**SnapATAC2.** SnapATAC2 (version 2.3.1) is a Python package developed in this study. We used the 'snapatac2.tl.spectral' function to generate the lower-dimensional embedding of the data. The output dimension was set to 30.

### Benchmarking metrics

To assess the quality of cell embeddings produced by various methods, we used a range of metrics: ARI, AMI, cell-type ASW and graph cLISI[43]. For batch effect removal analysis specifically, additional metrics were included: batch ASW, $k$-nearest-neighbor ($k$-NN) graph connectivity[43], graph iLISI[43], $k$-NN batch effect test (kBET)[43] and isolated label ASW.

To aggregate these individual metrics into a unified score, we first normalized each metric using min−max scaling, which involved subtracting the minimum value from each metric and then dividing it by the range. We then calculated the mean of these scaled metrics to derive an overall performance score for each method.

In the context of batch correction benchmarks, we categorized the metrics into two distinct groups: bio-conservation metrics and batch correction metrics. The bio-conservation group consists of ARI, AMI, ASW, graph cLISI and isolated label ASW. In contrast, the batch correction group included batch ASW, $k$-NN graph connectivity, graph iLISI and kBET. To calculate the overall performance score, denoted as $S_{overall}$, for each method, we took a weighted mean of the batch correction score, $S_{batch}$, and the bio-conservation score, $S_{bio}$, according to the equation: $S_{overall} = 0.4 \times S_{batch} + 0.6 \times S_{bio}$.

**ARI.** The ARI metric quantifies the degree of similarity between two different clusterings, accounting for both correct overlaps and disagreements. We generated a $k$-NN graph from cell embeddings with $k$ set at 50. Using this graph, we applied the Leiden algorithm[34] to obtain cell clusters. Given that the number of cell types in our benchmarking datasets is known, we fine-tuned the Leiden algorithm's resolution parameter between 0.1 and 3.0 in increments of 0.1 to match the actual number of clusters. Subsequently, we used ARI to evaluate the congruence between these Leiden clusters and the known cell-type labels. An ARI score of 0 indicates random labeling, while 1 represents a perfect match. We used the scikit-learn (v1.3.0) implementation for ARI calculations.

**AMI.** Like ARI, AMI also measures the similarity between two clusterings but is more effective when the reference clustering is imbalanced or contains small clusters[60]. The procedure for generating clusters and comparing them with cell-type labels mirrors that of ARI. The AMI scores range from 0 (random labeling) to 1 (perfect match) and were calculated using the scikit-learn (v1.3.0) package.

**ASW.** The ASW metric quantifies the degree of separation between clusters by averaging the silhouette widths across all cells. ASW values range from −1 to 1, with higher scores signaling better-defined clusters. However, the effectiveness of the ASW metric can be influenced by the dimensionality and topology of the data. Different dimensionality reduction methods can generate embeddings with varying numbers of dimensions, which can, in turn, impact the silhouette width. Additionally, the 'curse of dimensionality' poses challenges, as distance metrics become less reliable in higher-dimensional spaces. Moreover, silhouette width is most effective for evaluating convex clusters, but the shape of the clusters can vary based on the dimensionality reduction method used. To mitigate these issues, we standardized the dimensionality of all embeddings by applying the UMAP algorithm to reduce them to three dimensions. This not only facilitates a more equitable comparison but also enhances the reliability of the silhouette width as a metric. Using the scib-metrics software (version 0.3.3), we calculated two variants of the ASW to evaluate both cell-type separation (cell-type ASW) and batch mixing (batch ASW).

**Graph LISI.** The graph LISI metric extends the LISI by incorporating integrated graph structures to measure both batch mixing (graph iLISI) and cell-type separation (graph cLISI). LISI scores were computed using neighborhood lists from integrated $k$-NN graphs. The metric leverages the inverse Simpson's index to evaluate the diversity of cells within a neighborhood. We used the scIB (v0.3.3) package for these calculations.

**kBET.** The kBET algorithm tests if the label composition within a $k$-nearest neighborhood reflects the overall label composition. We used $k$-NN graphs with $k$ set at 50 for this purpose. The test was applied to a random subset of cells, and the rejection rate across all tested neighborhoods was summarized. kBET scores were computed using the scIB (v0.3.3) package.

**Isolated label ASW.** This metric specifically assesses how well data integration methods handle cell identity labels that are less commonly shared across batches. It calculates the ASW between isolated and non-isolated labels within the cell embedding, scaling the score between 0 and 1. The final score is the mean isolated score for all such labels, providing an evaluation of how well these less common labels are separated from other cell identities. Isolated label ASW calculations were performed using the scIB (v0.3.3) package.

### Scalability of scATAC-seq dimension reduction methods

To establish benchmarking datasets, we initially drew random cell samples from the Zemke_human dataset in varying numbers, ranging from 5,000 to 200,000 cells. From these samples, we constructed cell-by-bin matrices with a bin size of 500 base pairs, omitting any bins that were devoid of data across all cells. Subsequently, we applied various dimensionality reduction methods to these matrices using their default parameters and recorded both the runtimes and peak memory usages, plotting these metrics against the cell count. The benchmarks were conducted on a Linux server utilizing four cores of a 2.6 GHz Intel Xeon Platinum 8358 CPU.

For neural network-based techniques like PeakVI, scBasset and SCALE, we conducted the experiments on an A100 GPU equipped with 40 GB of memory. Notably, the memory usage of these methods is influenced more by the number of features than by the number of cells, due to the use of mini-batch training. When the feature count exceeded 500,000, we encountered memory limitations on the GPU. To mitigate this, we capped the feature set at 500,000 and opted not to report memory usage metrics for these methods, as they aren't directly comparable to other techniques. For benchmarking, we used a consistent set of 10 epochs to gauge the average runtime per epoch. We then extrapolated this to calculate the total runtime for a typical 50 epochs, which is generally the minimum required for model convergence. It's important to clarify that the runtimes reported for these neural network methods exclude data preprocessing time, thus representing a lower limit on the actual time needed.

## Preparing scHi-C benchmarking datasets

We obtained preprocessed 4DN scHi-C datasets[8] and a single-nucleus methyl-3C sequencing dataset[44] from a prior study[18], including cell-level contact matrices and cell labels. These datasets were already formatted for compatibility with Higashi. Additionally, we converted these datasets to formats suitable for input into scHiCluster and SnapATAC2.

## Comparing dimensionality reduction methods on scHi-C data

We used the prepared benchmarking datasets to evaluate various dimensionality reduction techniques. This benchmarking approach is analogous to the one used for scATAC-seq, as described earlier. Detailed operational specifics for each method are provided below.

**SnapATAC2.** We began by converting each cell's square region-by-region contact map into a vector. These vectors were then used to construct a sparse matrix representing all cells' contact maps. We used the 'snapatac2.pp.select_features' function to identify the top 500,000 features based on total counts. The method showed little sensitivity to the number of features selected. Finally, we used the 'snapatac2.tl.spectral' function to create a lower-dimensional embedding, setting the cell embedding dimension to 30.

**Higashi.** Higashi (GitHub: 392da1d9cd7208aef0e8f6f7b1192a-5aa0265ed2) is a Python package for analyzing scHi-C data. We followed the instructions in the Higashi GitHub repository to generate the lower-dimensional embedding of the data. The dimensionality of the cell embeddings was set to 30.

**scHiCluster.** scHiCluster (version 1.3.2) is a Python package for analyzing scHi-C data. We followed the instructions in the scHiCluster GitHub repository to generate the lower-dimensional embedding of the data. The dimensionality of the cell embeddings was set to 30.

**SCANPY (PCA).** For the SCANPY (PCA) method, we initially transformed each cell's square contact map into a vector and then constructed a sparse matrix, just like with SnapATAC2. The top 500,000 features with the highest total counts were selected. We used the 'scanpy.pp.normalize_total' and 'scanpy.pp.log1p' functions for data preprocessing. Lastly, we applied the 'scanpy.tl.pca' function to generate the lower-dimensional embedding. The initial output dimension was set to 30 and we used the elbow method to select the number of components retained for downstream analysis.

## Preparing scRNA-seq benchmarking datasets

We sourced five scRNA-seq datasets from a prior benchmarking study[35]. These datasets contain cell-by-gene count matrices and cell labels, and had already undergone preprocessing to eliminate low-quality cells,

## Comparing dimensionality reduction methods on scRNA-seq data

We utilized the curated benchmarking datasets to assess a range of dimensionality reduction techniques. The approach mirrors the one taken for scATAC-seq benchmarking, with a notable exception: before applying dimensionality reduction methods, we used the 'scanpy. pp.highly_variable_genes' function to identify the top 5,000 highly variable genes ('n_top_genes = 5,000'). Below are detailed explanations of the methods used.

**SnapATAC2.** To begin, we normalized the data using the 'scanpy.pp. normalize_total' and 'scanpy.pp.log1p' functions. Following this, the 'snapatac2.tl.spectral' function was utilized to create a lower-dimensional representation of the dataset. The cell embedding dimension was set to 30.

**SCANPY.** SCANPY (version 1.9.5) is a Python package for analyzing scRNA-seq data. To generate the lower-dimensional embedding of

the data, we first applied the 'scanpy.pp.normalize_total' and 'scanpy. pp.log1p' functions to preprocess the data. The data were then scaled using 'scanpy.pp.scale' with 'max_value = 10' and inputed to the 'scanpy.tl.pca' function to get lower-dimensional embedding. The initial output dimension was set to 30 and we used the elbow method to select the number of components retained for downstream analysis.

**scvi-tools.** scvi-tools (version 1.0.3) is a Python package for analyzing scRNA-seq data. We followed the instructions in the scvi-tools GitHub repository to generate the lower-dimensional embedding of the data, setting the dimensionality of the latent variable to 30.

## Preparing single-cell multiome benchmarking datasets

We obtained a paired ATAC and gene expression dataset of cryopreserved human PBMCs from the 10x Genomics website. Cell labels were annotated based on a previously published single-cell atlas of human PBMCs[4]. We used the 'Seurat::FindTransferAnchors' and 'Seurat::MapQuery' functions to map cell labels from the reference dataset to the 10x dataset, using 'spca' as the reference reduction method and 'wnn.umap' as the reduction model. Cells were then filtered based on a minimum threshold of 200 detected genes, 5,000 ATAC fragments and a TSS enrichment score of at least 10. Doublets were removed using 'snapatac2.pp.scrublet'. Cell-by-gene and cell-by-bin matrices were constructed for scRNA-seq and scATAC-seq data, respectively. The bin size was set to 500 bp, and the top 500,000 most accessible bins were selected using the 'snapatac2.pp.select_features' function. The finalized dataset contained 9,181 cells.

As an additional benchmarking dataset, we downloaded a Paired-Tag dataset from a study on the mouse frontal cortex[53], which simultaneously measures H3K27me3 histone modification and gene expression at single-cell resolution. We obtained the cell-by-gene matrix from the publication and created the cell-by-bin matrix using the fragment files provided by the authors, with a bin size of 5 kb as recommended in the original paper. The top 100,000 most accessible bins were selected using the 'snapatac2.pp.select_features' function, and cell labels were sourced from the original paper. The final dataset in this case comprised 10,180 cells.

## Comparing cell embedding methods on single-cell multiome data

We used the two curated single-cell multiome datasets to evaluate four methods designed for joint cell embedding across multiple data modalities. For gene expression data, the top 3,000 highly variable genes were selected, while all features were included for ATAC or histone modification data. Due to memory limitations, MOFA+ used a maximum of 200,000 features for ATAC data. The accuracy of these dimensionality reduction methods was assessed using four distinct evaluation metrics, which are elaborated in 'Benchmarking metrics'. A comprehensive elucidation on the operational specifics of each method is provided below.

**SnapATAC2.** For normalization of gene expression data, we used the 'scanpy.pp.normalize_total' and 'scanpy.pp.log1p' functions. Subsequently, the 'snapatac2.tl.multi_spectral' function was applied to jointly reduce the dimensionality of both the gene expression and ATAC data, setting the output dimensionality to 30.

**MIRA.** MIRA (version 2.1.0) is a Python package focused on analyzing dynamic gene regulation processes in single-cell multi-omics datasets. To generate a joint embedding, we initially conducted topic modeling on each modality using 'mira.topics.make_model' and selected the number of topics via 'mira.topics.gradient_tune' and the elbow method. The joint representation was then obtained using 'mira.utils. make_joint_representation'.

**MOFA+.** MOFA+ (mofapy2, version 0.7.0) uses a computationally efficient variational inference to create a low-dimensional representation of multimodal data. We used MOFA+ through its Python implementation in the 'MUON' package[61]. The gene expression and chromatin data were preprocessed using 'scanpy.pp.normalize_total', 'scanpy.pp.log1p' and 'muon.atac.pp.tfidf' functions, respectively. The joint embedding was achieved using 'muon.tl.mofa' setting the 'n_factors' parameter to 30.

**Cobolt.** Cobolt (version 1.0.1) is another Python package designed for single-cell data analysis from joint-modality platforms, utilizing a multimodal variational autoencoder. The number of latent dimensions was set to 30, and a learning rate of 0.002 was applied, per recommendations found in the Cobolt GitHub repository.

### Benchmarking batch effect correction methods on scRNA-seq and scATAC-seq data

Several methods, such as scVI and PeakVI, incorporate batch correction during the dimensionality reduction phase. In contrast, other approaches like Harmony, Scanorama and FastMNN mitigate batch effects during a post-processing stage; SnapATAC2 falls under this category. We initially utilized SnapATAC2's dimensionality reduction algorithm to generate cell embeddings and subsequently applied various algorithms for batch effect correction. These outcomes were then compared with alternative methods that either directly address batch effects in the raw data based on batch labels or operate on the cell embeddings. Evaluation metrics, as detailed in 'Benchmarking metrics', were used to assess the performance of these various approaches. We leveraged the 'scib-pipeline'[43] to conduct this analysis on four distinct scRNA-seq datasets. Additionally, we included two scATAC-seq datasets, GSE194122 and Zemke_human, to specifically evaluate the performance of batch effect correction methods in scATAC-seq contexts.

### Assessing SnapATAC2's ability to detect rare cell types

To evaluate how effectively SnapATAC2 can identify rare cell types, we used synthetic human bone marrow datasets described earlier. Our focus was primarily on the CD8[+] T cell population, which we strategically downsampled to make up various proportions of the total cell count, ranging from a scant 0.5% up to 15%. The CD8[+] T cells were chosen for this experiment because their chromatin accessibility profiles are relatively similar to those of CD4[+] T cells in the dataset. This characteristic presents a meaningful challenge for accurately distinguishing between the two cell types. Subsequently, we applied a range of dimensionality reduction algorithms to these altered datasets. The effectiveness of each method was assessed by calculating the ASW for the CD8[+] T cell population based on their embeddings. Specifically, the silhouette width metric was used to quantify the separation between the CD8[+] T cells and their closest neighboring cell population, thereby providing an insight into each algorithm's capacity to differentiate this rare cell type effectively.

### Reporting summary

Further information on research design is available in the Nature Portfolio Reporting Summary linked to this article.

### Data availability

We processed various public scATAC-seq datasets for our benchmarking analysis, with the datasets listed in Table 1. These include: 10x Genomics scATAC-seq data for brain (5,000 cells) and PBMCs (10,000 cells), datasets from Chen et al., Ma et al. and Yao et al., available at http://ftp.cbi.pku.edu.cn/pub/glue-download/; Buenrostro et al. and synthetic scATAC-seq datasets, accessible on GitHub at https://github.com/pinellolab/scATAC-benchmarking/; human PBMC data from the Gene Expression Omnibus (GEO; GSE194122); human cerebral cortex data from GSE162170 (GEO);

data from both human and mouse primary motor cortex, available from GSE229169 (GEO). Additionally, we utilized scHi-C datasets downloaded from Google Drive at[18] https://drive.google.com/drive/u/0/folders/1j7ffz96kv_Ft3hicu2DRBmfcje0RA1sc/ and scRNA-seq datasets obtained through the 'DuoClustering2018' R package, which can be found on Bioconductor. Our study also includes single-cell multiome data for human PBMCs, downloaded from 10x Genomics at https://www.10xgenomics.com/resources/datasets/pbmc-from-a-healthy-donor-granulocytes-removed-through-cell-sorting-10-k-1-standard-2-0-0/, as well as Paired-Tag data in mouse frontal cortex from GSE224560 (GEO). All the processed datasets generated in our study are made available as Anndata objects and can be accessed at https://osf.io/hfs2v/. Source data are provided with this paper.

### Code availability

The source code of SnapATAC2 can be accessed at https://github.com/kaizhang/SnapATAC2/. The source code for reproducing the benchmarks in this project can be accessed at https://github.com/kaizhang/single-cell-benchmark/.

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

### Acknowledgements

We thank Y. Zhang (Westlake University), S. Zu (UCSD) and Y. E. Li (UCSD) and other members of the B.R. Lab for valuable discussion and feedback on earlier versions of the manuscript. This work was supported by the Ludwig Institute for Cancer Research (to B.R.), and National Institutes of Health grants (U01HG012059, UM1HG011585, RF1MH128838, UM1MH130994, R24AG073198, U01MH114828, R56AG069107, U54AG079758, U01MH121282, U19MH114831 and R01 EY031663, to B.R.). The funders had no role in study design, data collection and analysis, decision to publish or preparation of the manuscript.

### Author contributions

Conceptualization: K.Z. and B.R.; methodology: K.Z.; software: K.Z.; investigation: K.Z. and B.R.; data curation: K.Z., N.R.Z. and E.J.A.; writing—original draft: K.Z. and B.R.; writing—review and editing: K.Z., N.R.Z., E.J.A. and B.R.; funding acquisition: B.R.

### Competing interests

B.R. is a cofounder of Epigenome Technologies, and a cofounder and consultant of Arima Genomics. The remaining authors declare no competing interests.

### Additional information

**Extended data** is available for this paper at https://doi.org/10.1038/s41592-023-02139-9.

**Correspondence and requests for materials** should be addressed to Bing Ren.

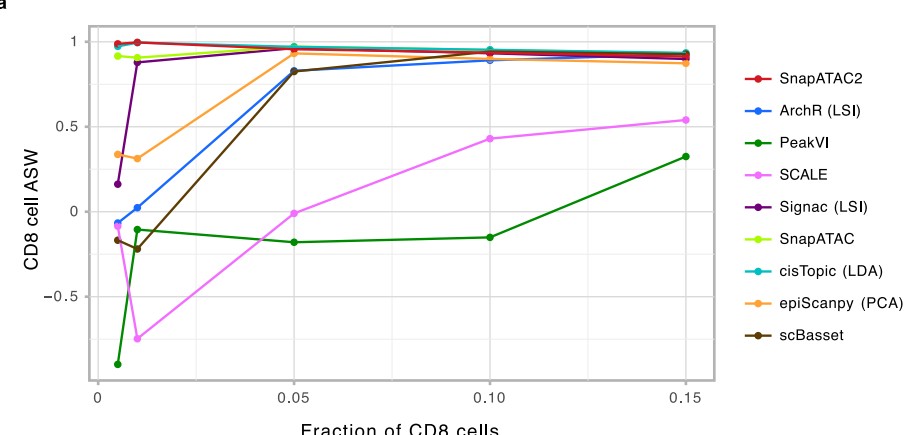

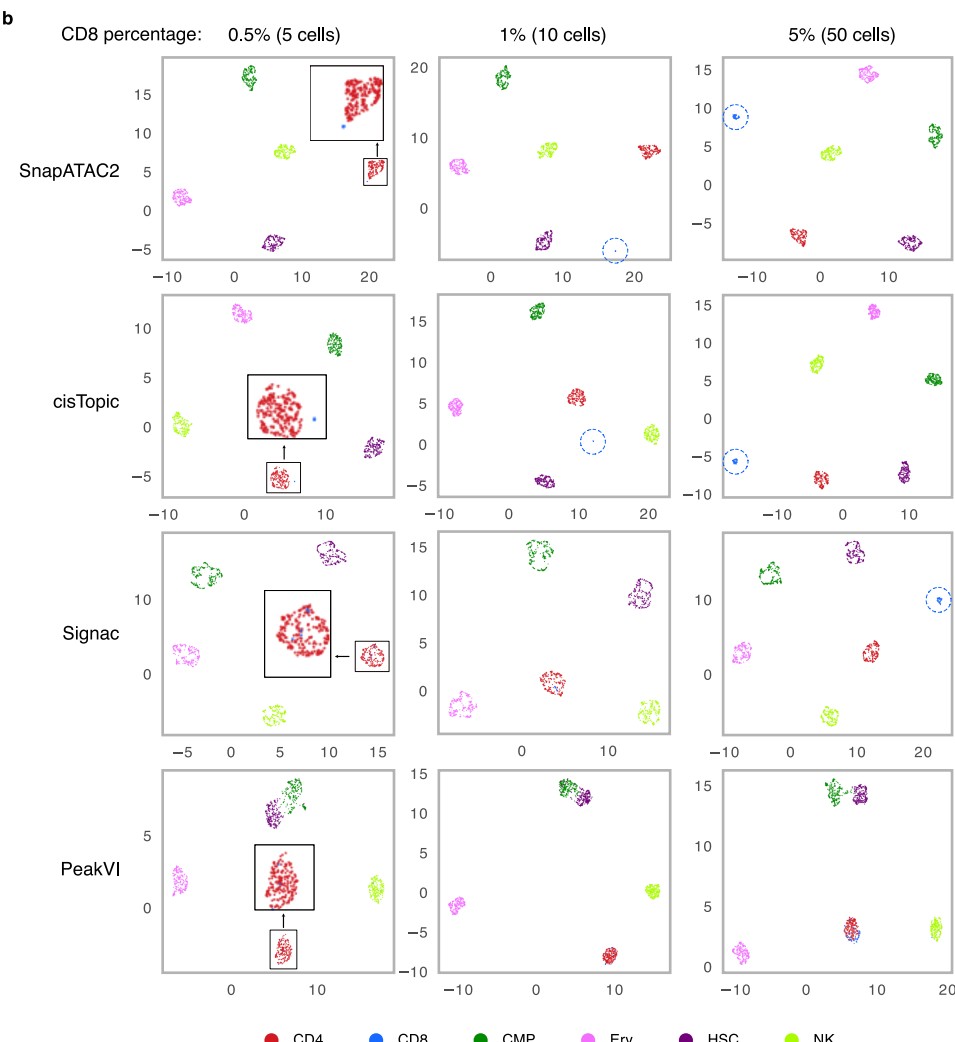

**Extended Data Fig. 1 | SnapATAC2 excels at identifying rare cell types. a**, Line plot showing the average silhouette scores of CD8⁺ T cells (Y-axis) as a function of the fraction of CD8⁺ T cells in the dataset (X-axis) across nine dimensionality reduction methods. **b**, UMAP visualization of the embeddings produced by selected methods on the datasets with varying fractions (0.5%, 1%, 5%) of CD8⁺ T cells.

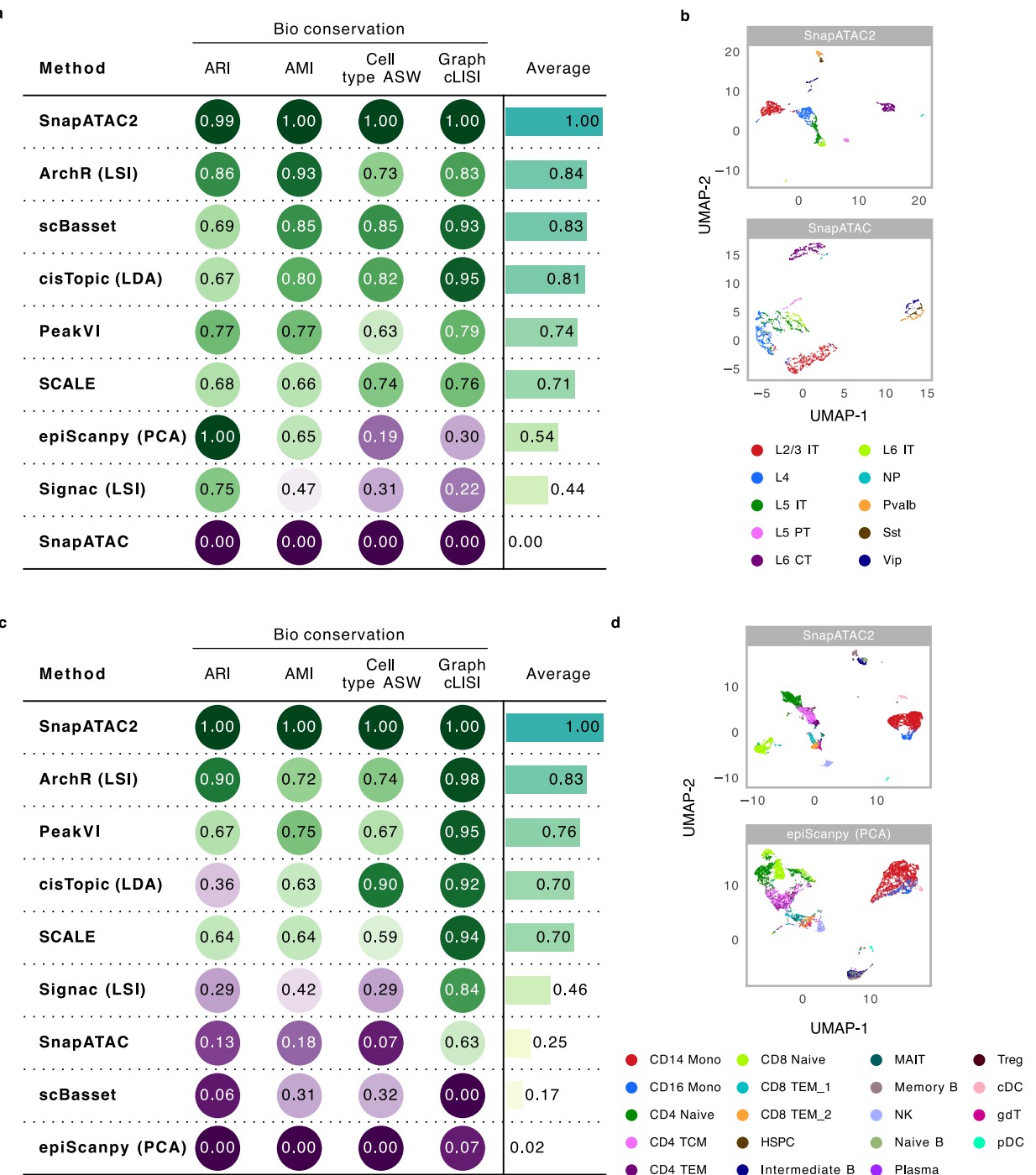

**Extended Data Fig. 2 | Benchmarking of dimensionality reduction methods on 10× Brain 5k and PBMC 10k datasets. a,c,** Tables displaying normalized scores (0–1 range) of four metrics used to evaluate each method's bio-conservation on the 10× Brain 5k (**a**) and PBMC 10k (**c**) datasets. A score of 1 indicates optimal performance. See Methods for metric details. **b,d,** UMAP visualizations of the embeddings generated by the best performing method and the worst performing method on the 10× Brain 5k (**b**) and PBMC 10k (**d**) datasets. Cells are color-coded by cell type labels.

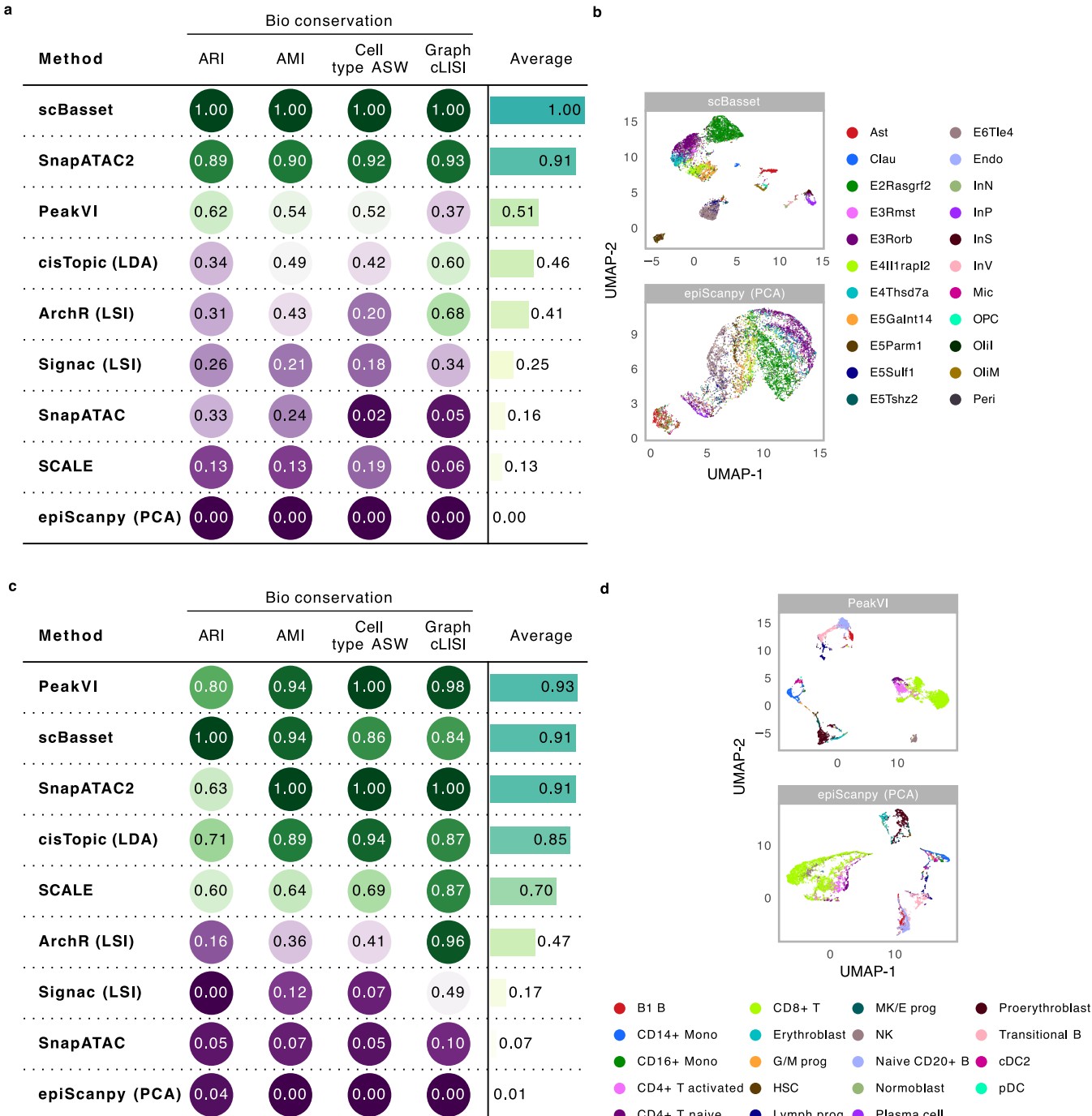

**Extended Data Fig. 3 | Benchmarking of dimensionality reduction methods on the Chen et al. and GSE194122 datasets. a,c** Tables displaying normalized scores (0–1 range) of four metrics used to evaluate each method's bio-conservation on the Chen et al. (**a**) and GSE194122 (**c**) datasets. A score of 1 indicates optimal performance. See Methods for metric details. **b,d**, UMAP visualizations of the embeddings generated by the best performing method and the worst performing method on the Chen et al. (**b**) and GSE194122 (**d**) datasets. Cells are color-coded by cell type labels.

**a**

| Method | Bio conservation | | | | Average |
|---|---|---|---|---|---|
| | ARI | AMI | Cell type ASW | Graph cLISI | |
| **SnapATAC2** | 1.00 | 1.00 | 1.00 | 1.00 | 1.00 |
| **PeakVI** | 0.82 | 0.93 | 0.83 | 0.99 | 0.89 |
| **cisTopic (LDA)** | 0.79 | 0.93 | 0.70 | 0.94 | 0.84 |
| **ArchR (LSI)** | 0.69 | 0.85 | 0.54 | 0.98 | 0.76 |
| **SnapATAC** | 0.68 | 0.82 | 0.56 | 0.92 | 0.74 |
| **epiScanpy (PCA)** | 0.65 | 0.82 | 0.56 | 0.93 | 0.74 |
| **SCALE** | 0.60 | 0.71 | 0.50 | 0.79 | 0.65 |
| **Signac (LSI)** | 0.47 | 0.68 | 0.20 | 0.82 | 0.54 |
| **scBasset** | 0.00 | 0.00 | 0.00 | 0.00 | 0.00 |

**b**

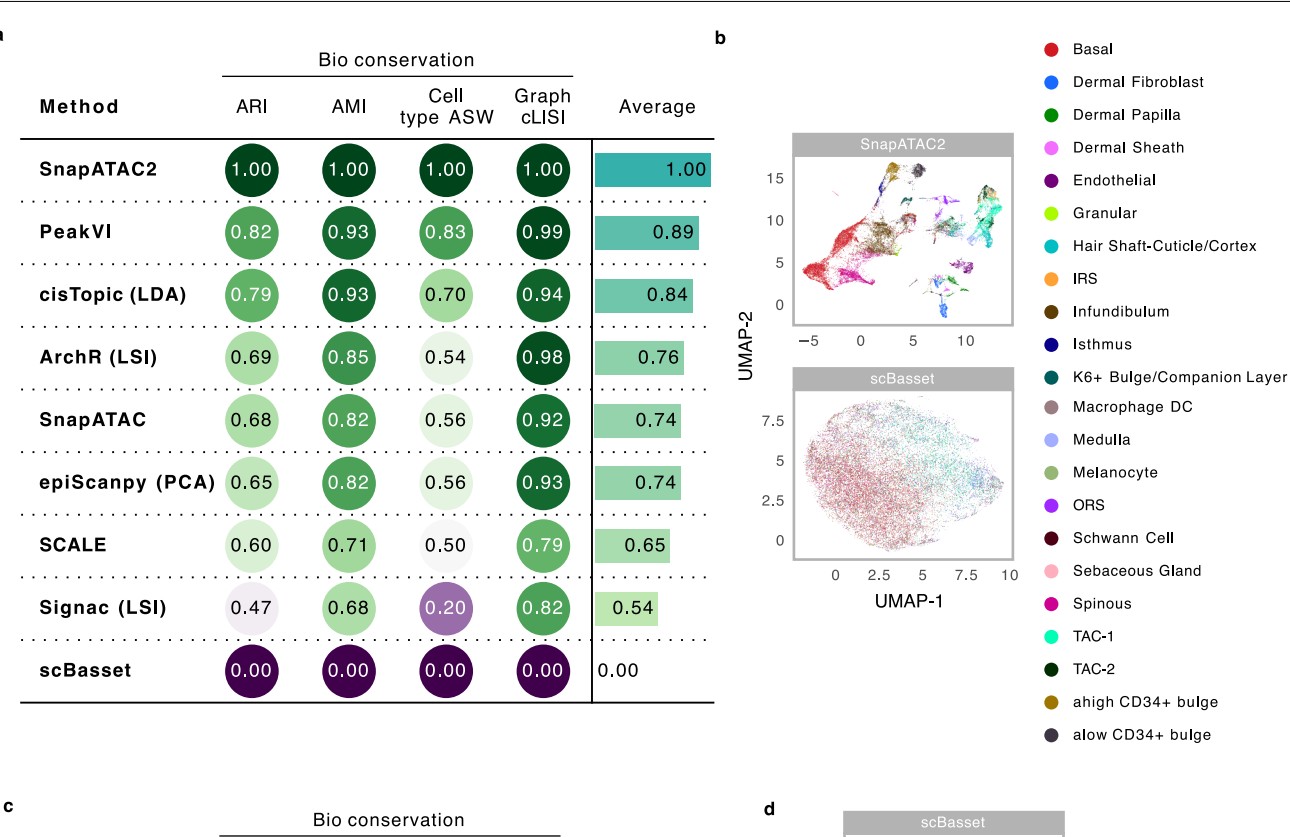

**c**

| Method | Bio conservation | | | | Average |
|---|---|---|---|---|---|
| | ARI | AMI | Cell type ASW | Graph cLISI | |
| **scBasset** | 0.98 | 0.96 | 1.00 | 1.00 | 0.99 |
| **SnapATAC2** | 1.00 | 1.00 | 0.62 | 0.99 | 0.90 |
| **PeakVI** | 0.61 | 0.62 | 0.75 | 0.58 | 0.64 |
| **cisTopic (LDA)** | 0.84 | 0.88 | 0.53 | 0.08 | 0.58 |
| **epiScanpy (PCA)** | 0.55 | 0.36 | 0.53 | 0.48 | 0.48 |
| **ArchR (LSI)** | 0.55 | 0.27 | 0.45 | 0.58 | 0.46 |
| **SnapATAC** | 0.56 | 0.34 | 0.41 | 0.48 | 0.45 |
| **SCALE** | 0.58 | 0.27 | 0.30 | 0.00 | 0.29 |
| **Signac (LSI)** | 0.00 | 0.00 | 0.00 | 0.22 | 0.06 |

**d**

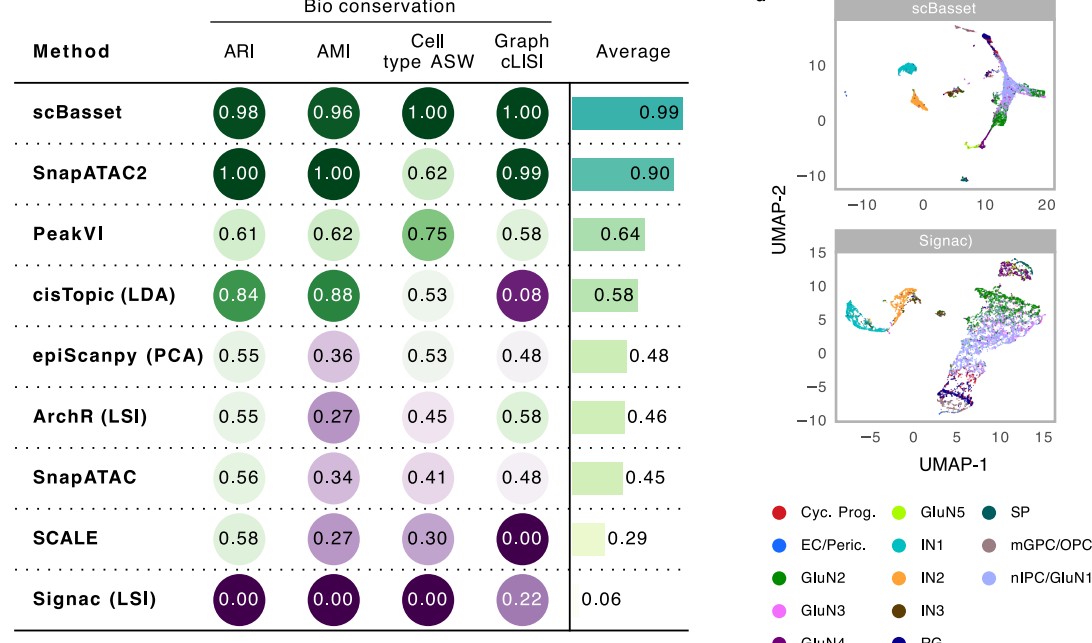

**Extended Data Fig. 4 | Benchmarking of dimensionality reduction methods on the Ma et al. and Trevino et al. datasets. a,c**, Tables displaying normalized scores (0–1 range) of four metrics used to evaluate each method's bio-conservation on the Ma et al. (**a**) and Trevino et al. (**c**) datasets. A score of 1 indicates optimal performance. See Methods for metric details. **b,d**, UMAP visualizations of the embeddings generated by the best performing method and the worst performing method on the Ma et al. (**b**) and Trevino et al. (**d**) datasets. Cells are color-coded by cell type labels.

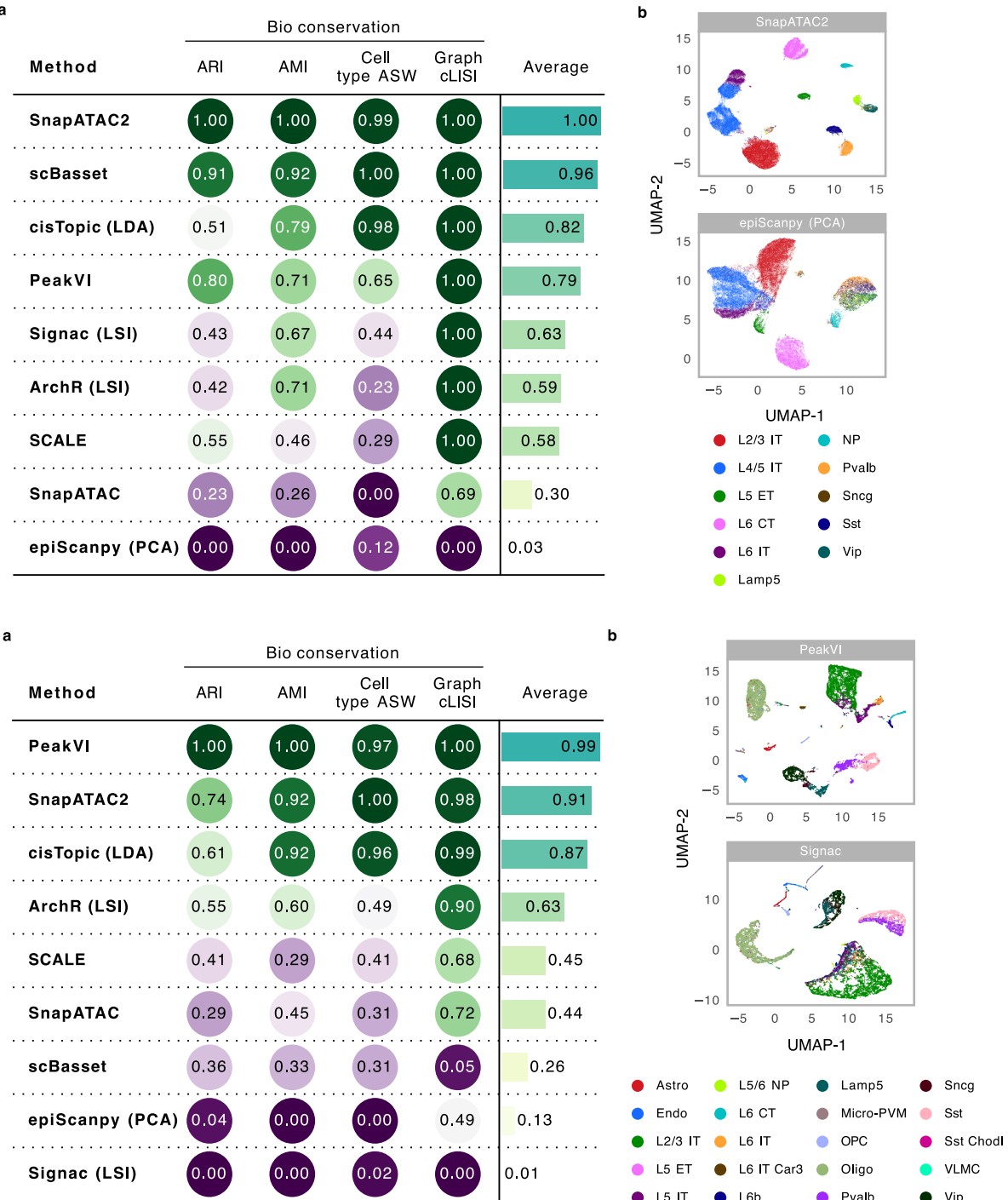

**Extended Data Fig. 5 | Benchmarking of dimensionality reduction methods on the Yao et al. and Zemke et al. human datasets. a,c,** Tables displaying normalized scores (0–1 range) of four metrics used to evaluate each method's bio-conservation on the Yao et al. (**a**) and Zemke et al. human (**c**) datasets. A score of 1 indicates optimal performance. See Methods for metric details. **b,d,** UMAP visualizations of the embeddings generated by the best performing method and the worst performing method on the Yao et al. (**b**) and Zemke et al. human (**d**) datasets. Cells are color-coded by cell type labels.

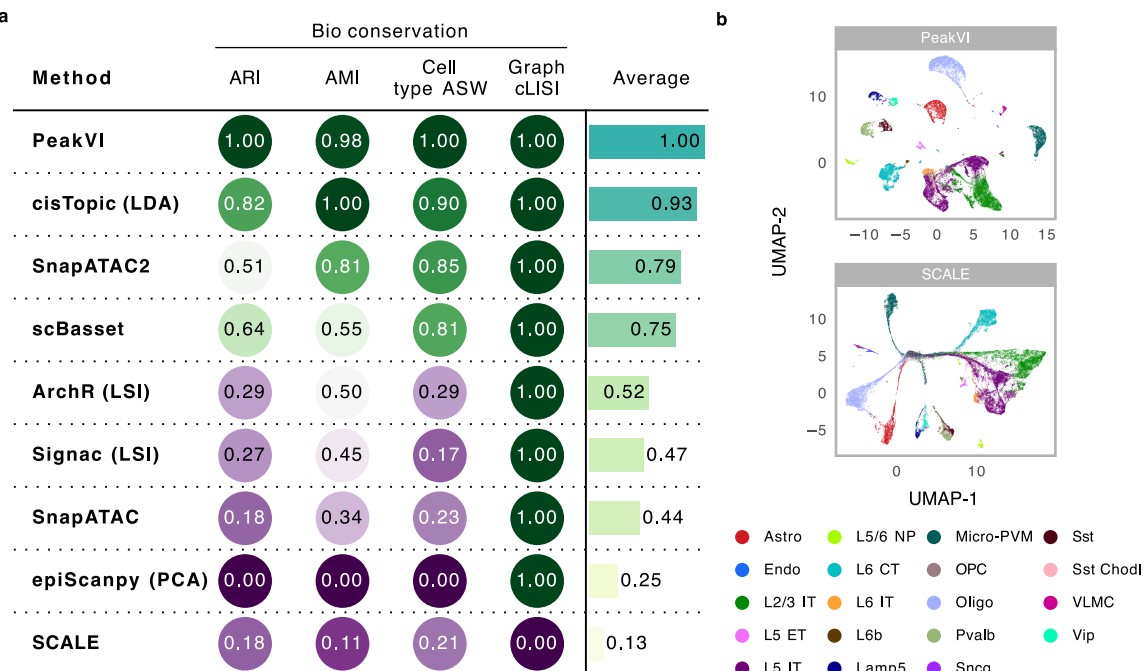

**Extended Data Fig. 6 | Benchmarking of dimensionality reduction methods on the Zemke et al. mouse dataset. a**, Table displaying normalized scores (0–1 range) of four metrics used to evaluate each method's bio-conservation. A score of 1 indicates optimal performance. See Methods for metric details. **b**, UMAP visualization of the embeddings generated by the best performing method and the worst performing method. Cells are color-coded by cell type labels.

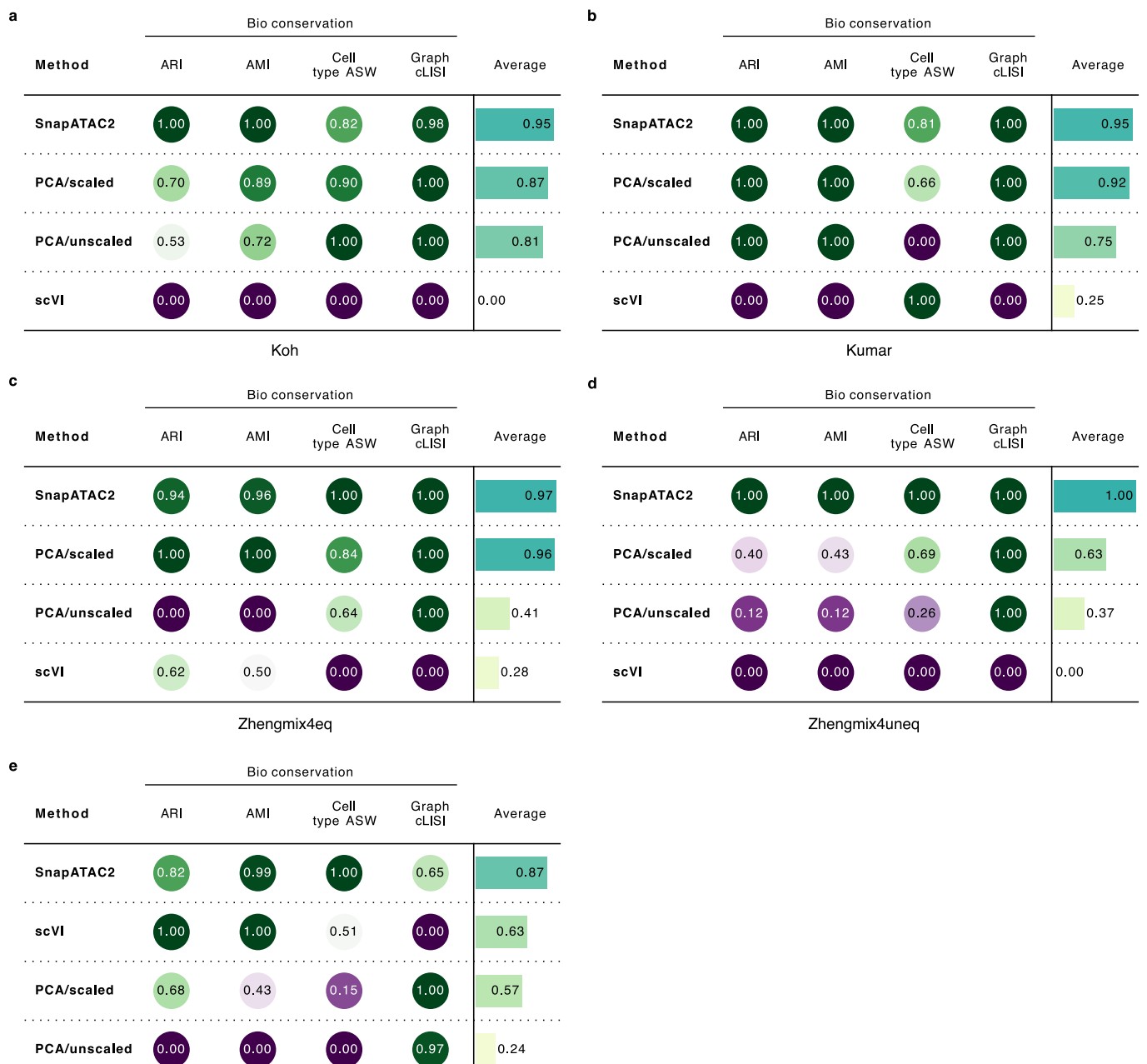

**Extended Data Fig. 7 | SnapATAC2 demonstrates superior performance over other methods on scRNA-seq datasets.** Table displaying normalized scores (0–1 range) of four metrics used to evaluate each method's bio-conservation on five datasets, including Koh (**a**), Kumar (**b**), Zhengmix4eq (**c**), Zhengmix4uneq (**d**), and Zhengmix8eq (**e**).

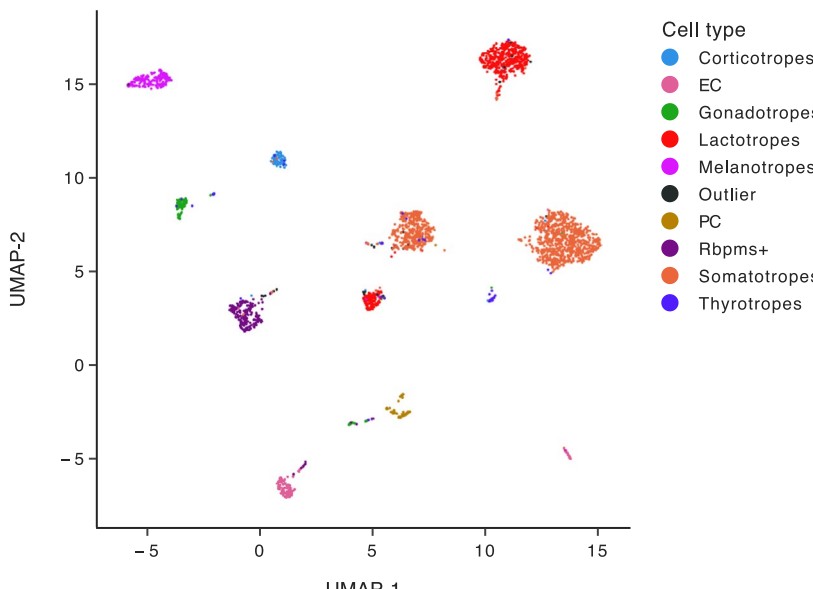

**Extended Data Fig. 8 | SnapATAC2 unveils fine-grained cellular heterogeneity in single-cell DNA methylation data from Ruf-Zamojski et al.** UMAP visualization of cell embedding generated by SnapATAC2. Cells are colored by cell type labels.

**a**

| Method | scRNA−seq dataset | | | | Overall |
| | immune_human | immune_human_mouse | lung_atlas | pancreas | |
| --- | --- | --- | --- | --- | --- |
| scvi | 0.74 | 0.77 | 0.67 | 0.69 | 0.72 |
| fastmnn | 0.68 | 0.69 | 0.66 | 0.69 | 0.68 |
| SnapATAC2/mnc | 0.64 | 0.62 | 0.63 | 0.77 | 0.67 |
| SnapATAC2/harmony | 0.72 | 0.46 | 0.60 | 0.84 | 0.66 |
| scanorama | 0.64 | 0.68 | 0.43 | 0.62 | 0.59 |
| bbknn | 0.57 | 0.58 | 0.42 | 0.67 | 0.56 |
| seuratrpca | 0.43 | 0.46 | 0.48 | 0.75 | 0.53 |
| harmony | 0.52 | 0.35 | 0.43 | 0.75 | 0.51 |
| combat | 0.34 | 0.55 | 0.44 | 0.55 | 0.47 |
| SnapATAC2/scanorama | 0.40 | 0.43 | 0.67 | 0.26 | 0.44 |

**b**

| Method | scATAC−seq dataset | | Overall |
| | GSE194122 | Zemke_2023_human | |
| --- | --- | --- | --- |
| PeakVI | 0.78 | 0.81 | 0.79 |
| SnapATAC2/mnc | 0.74 | 0.79 | 0.77 |
| cisTopic/harmony | 0.76 | 0.71 | 0.73 |
| SnapATAC2/scanorama | 0.72 | 0.70 | 0.71 |
| SnapATAC2/harmony | 0.71 | 0.64 | 0.67 |
| cisTopic/scanorama | 0.70 | 0.61 | 0.65 |
| cisTopic/mnc | 0.66 | 0.62 | 0.64 |
| ArchR (log(TF-IDF))/scanorama | 0.49 | 0.72 | 0.61 |
| ArchR (log(TF-IDF))/harmony | 0.50 | 0.63 | 0.56 |
| SnapATAC/scanorama | 0.56 | 0.57 | 0.56 |
| epiScanpy (PCA)/harmony | 0.65 | 0.47 | 0.56 |
| Signac (log(TF-IDF))/harmony | 0.52 | 0.58 | 0.55 |
| Signac (log(TF-IDF))/scanorama | 0.49 | 0.59 | 0.54 |
| SnapATAC/mnc | 0.52 | 0.56 | 0.54 |
| SnapATAC/harmony | 0.50 | 0.52 | 0.51 |
| ArchR (log(TF-IDF))/mnc | 0.39 | 0.54 | 0.46 |
| scBasset | 0.54 | 0.31 | 0.43 |
| Signac (log(TF-IDF))/mnc | 0.37 | 0.48 | 0.42 |
| epiScanpy (PCA)/mnc | 0.41 | 0.39 | 0.40 |
| epiScanpy (PCA)/scanorama | 0.38 | 0.41 | 0.39 |

**Extended Data Fig. 9 | SnapATAC2 remains robust and reliable when processing datasets with batch effects.** Tables showing the aggregated scores for bio-conservation and batch correction metrics across different scRNA-seq datasets (**a**) and scATAC-seq datasets (**b**) for each method. For more details, see Supplementary Tables 1 and 2.

**a**

---

**Algorithm 1:** Matrix-free spectral embedding with cosine similarity

---

**Input** : Matrix $X \in \mathbb{R}^{n \times p}$ and an integer $k$
**Output:** eigenvalues $\lambda \in \mathbb{R}^{k \times 1}$ and eigenvectors $V \in \mathbb{R}^{n \times k}$

1 Normalize each row in $X$ to unit norm
2 $D \leftarrow diag(X(X^T \mathbf{1}) - \mathbf{1})$
3 $\tilde{X} \leftarrow D^{-1/2} X$
4 Define the vector transformation function:
   $F(v) = \tilde{X}(\tilde{X}^T v) - D^{-1} v$
5 Plug in the linear operator $F$ into the Lanczos algorithm to get the top $k$-largest eigenvalues $\lambda$ and corresponding eigenvectors $V$.

---

**b**

---

**Algorithm 2:** Nystrom

---

**Input** : $Z \in \mathbb{R}^{n \times p}$ containing $n$ training samples
   $Y \in \mathbb{R}^{m \times p}$ containing $m$ out-of-bag samples
**Output:** eigenvalues $\Lambda \in \mathbb{R}^{k \times k}$ and eigenvectors $U \in \mathbb{R}^{m \times k}$

1 Perform spectral embedding on $Z$ to get eigenvalues $\lambda$ and eigenvectors $V$
2 $\Lambda \leftarrow diag(\lambda)$
3 Normalize rows in $Z$ and $Y$ to unit norm
4 $D \leftarrow diag(Z(Z^T \mathbf{1}) - \mathbf{1})$
5 $B \leftarrow D^{-\frac{1}{2}} V \Lambda^{-1}$
6 $Q \leftarrow Y Z^T B$
7 $\hat{D} \leftarrow diag(Q\Lambda Q^T \mathbf{1})$
8 $U \leftarrow \hat{D}^{-\frac{1}{2}} Q$
9 Update $U, \Lambda \leftarrow \text{Orthogonalization}(U, \Lambda)$

---

**c**

---

**Algorithm 3:** Orthogonalization

---

**Input** : eigenvectors $U \in \mathbb{R}^{m \times k}$ and eigenvalues $\Lambda \in \mathbb{R}^{k \times k}$
**Output:** orthogonalized $\tilde{U}$ and $\tilde{\Lambda}$

1 $P \leftarrow U^T U$
2 eigen-decomposition: $P = V\Sigma V^T$
3 $B \leftarrow \Sigma^{1/2} V^T \Lambda V \Sigma^{1/2}$
4 eigen-decomposition: $B = \tilde{V}\tilde{\Lambda}\tilde{V}^T$. Reorder the eigenvalues in $\tilde{\Lambda}$ and according eigenvalues if necessary.
5 $\tilde{U} \leftarrow UV\Sigma^{-1/2}\tilde{V}$

---

**d**

---

**Algorithm 4:** Matrix-free multi-view spectral embedding with cosine similarity

---

**Input** : Matrices $X_1, X_2, ..., X_k \in \mathbb{R}^{n \times p_k}$
   Non-negative weights $\lambda_1, \lambda_2, ..., \lambda_k$
   An integer $k$
**Output:** eigenvalues $\lambda \in \mathbb{R}^{k \times 1}$ and eigenvectors $V \in \mathbb{R}^{n \times k}$

1 **for** $i \leftarrow 1$ **to** $k$ **do**
2  Perform row-wise $L_2$-norm normalization on $X_i$
3  $\hat{X}_i \leftarrow$ random sample of rows of $X_i$
4  $w_i \leftarrow \frac{\lambda_i}{\|\hat{X}_i \hat{X}_i^T - I\|}$
5 **end for**
6 $S \leftarrow \sum_i w_i$
7 **for** $i \leftarrow 1$ **to** $k$ **do**
8  Update $X_i \leftarrow \sqrt{\frac{w_i}{S}} X_i$
9 **end for**
10 $X \leftarrow$ horizontally concatenate $X_1, X_2, ..., X_k$
11 $D \leftarrow diag(X(X^T \mathbf{1}) - \mathbf{1})$
12 $\tilde{X} \leftarrow D^{-1/2} X$
13 Define the vector transformation function:
   $F(v) = \tilde{X}(\tilde{X}^T v) - D^{-1} v$
14 Plug in the linear operator $F$ into the Lanczos algorithm to get the top $k$-largest eigenvalues $\lambda$ and corresponding eigenvectors $V$.

---

**Extended Data Fig. 10 | The pseudocodes of various algorithms used in this study. a**, The pseudocode of the matrix-free spectral embedding algorithm. **b**, The pseudocode of the Nyström algorithm for performing the out-of-sample embedding. **c**, The pseudocode for performing orthogonalization on the eigenvectors produced by the Nyström algorithm. **d**, The pseudocode of the matrix-free multi-view spectral embedding algorithm.

# Reporting Summary

## Statistics

For all statistical analyses, confirm that the following items are present in the figure legend, table legend, main text, or Methods section.

| n/a | Confirmed | |
|---|---|---|
| ☐ | ☒ | The exact sample size (*n*) for each experimental group/condition, given as a discrete number and unit of measurement |
| ☐ | ☒ | A statement on whether measurements were taken from distinct samples or whether the same sample was measured repeatedly |
| ☐ | ☒ | The statistical test(s) used AND whether they are one- or two-sided *Only common tests should be described solely by name; describe more complex techniques in the Methods section.* |
| ☐ | ☒ | A description of all covariates tested |
| ☐ | ☒ | A description of any assumptions or corrections, such as tests of normality and adjustment for multiple comparisons |
| ☐ | ☒ | A full description of the statistical parameters including central tendency (e.g. means) or other basic estimates (e.g. regression coefficient) AND variation (e.g. standard deviation) or associated estimates of uncertainty (e.g. confidence intervals) |
| ☐ | ☒ | For null hypothesis testing, the test statistic (e.g. *F*, *t*, *r*) with confidence intervals, effect sizes, degrees of freedom and *P* value noted *Give P values as exact values whenever suitable.* |
| ☒ | ☐ | For Bayesian analysis, information on the choice of priors and Markov chain Monte Carlo settings |
| ☒ | ☐ | For hierarchical and complex designs, identification of the appropriate level for tests and full reporting of outcomes |
| ☒ | ☐ | Estimates of effect sizes (e.g. Cohen's *d*, Pearson's *r*), indicating how they were calculated |

*Our web collection on statistics for biologists contains articles on many of the points above.*

## Software and code

Policy information about availability of computer code

| Data collection | DuoClustering2018 (version 1.20.0) |
|---|---|
| Data analysis | We have used the following software packages for data analysis: SnapATAC2 (version 2.3.1), ArchR (version 1.0.1), Signac (version 1.6), Episcanpy (version 0.4.0), SCALE (version 1.1.2), PeakVI (version 0.19.0), scBasset (github: c15bec3a73fa1e04822db723338d234ca9d384ce), pycisTopic (github: 242c2a47aad475250f8abb2469a0e36085d6e460), SnapATAC (version 1.0), Higashi (github: 392da1d9cd7208aef0e8f6f7b1192a5aa0265ed2), scHiCluster (version 1.3.2), SCANPY (version 1.9.5), scvi-tools (version 1.0.3), mofapy2 (version 0.7.0), MIRA (version 2.1.0), Cobolt (version 1.0.1), scikit-learn (version 1.3.0), scib-metrics (version 0.3.3), scIB (version 0.3.3) Custom code used for benchmark study: https://github.com/kaizhang/single-cell-benchmark. |

For manuscripts utilizing custom algorithms or software that are central to the research but not yet described in published literature, software must be made available to editors and reviewers. We strongly encourage code deposition in a community repository (e.g. GitHub). See the Nature Portfolio guidelines for submitting code & software for further information.

## Data

Policy information about availability of data

All manuscripts must include a data availability statement. This statement should provide the following information, where applicable:
- Accession codes, unique identifiers, or web links for publicly available datasets
- A description of any restrictions on data availability
- For clinical datasets or third party data, please ensure that the statement adheres to our policy

We processed various public scATAC-seq datasets for our benchmarking analysis, with the datasets listed in Table 1. These include: 10x Genomics scATAC-seq data for brain (5k cells) and peripheral blood mononuclear cells (PBMCs, 10k cells), datasets from Chen et al., Ma et al., and Yao et al., available at this FTP server: http://ftp.cbi.pku.edu.cn/pub/glue-download/; Buenrostro2018 and synthetic scATAC-seq datasets, accessible on GitHub: https://github.com/pinellolab/scATAC-benchmarking; Human PBMCs data from GSE194122 (Gene Expression Omnibus (GEO)); Human cerebral cortex data from GSE162170 (GEO); Data from both human and mouse primary motor cortex, available from GSE229169 (GEO). Additionally, we utilized scHi-C datasets downloaded from this Google Drive link: https://drive.google.com/drive/u/0/folders/1j7ffz96kv_Ft3hicu2DRBmfcjeORA1sc, and scRNA-seq datasets obtained through the "DuoClustering2018" R package, which can be found on Bioconductor. Our study also includes single-cell multiome data for human PBMCs, downloaded from 10x Genomics: https://www.10xgenomics.com/resources/datasets/pbmc-from-a-healthy-donor-granulocytes-removed-through-cell-sorting-10-k-1-standard-2-0-0, as well as Paired-Tag data in mouse frontal cortex from GSE224560 (GEO). All the processed datasets generated in our study are made available as Anndata objects and can be accessed here: https://osf.io/hfs2v/.

## Human research participants

Policy information about studies involving human research participants and Sex and Gender in Research.

| | |
|---|---|
| Reporting on sex and gender | Not applicable |
| Population characteristics | Not applicable |
| Recruitment | Not applicable |
| Ethics oversight | Not applicable |

Note that full information on the approval of the study protocol must also be provided in the manuscript.

# Field-specific reporting

Please select the one below that is the best fit for your research. If you are not sure, read the appropriate sections before making your selection.

☒ Life sciences ☐ Behavioural & social sciences ☐ Ecological, evolutionary & environmental sciences

For a reference copy of the document with all sections, see nature.com/documents/nr-reporting-summary-flat.pdf

# Life sciences study design

All studies must disclose on these points even when the disclosure is negative.

| | |
|---|---|
| Sample size | The benchmark data were assembled to encompass a variety of experiment protocols, tissues, species, and data types, resulting in 20 unique datasets. When evaluating individual computational methods, we did not gather data from multiple runs since these methods exhibit minimal or no variation. Exceptions were made for benchmarking subsampling methods, where we used 9 samples, and for run time benchmarking, where 3 samples were taken. These sample sizes were selected to stabilize the variance. |
| Data exclusions | Data exclusions were carried out for the GSE194122 and Zemke_human datasets. Because in these cases, the data was generated using multiple donors or protocols, leading to pronounced batch effects. To ensure that our benchmark was not skewed by these batch effects, we opted to use only a subset of cells from these two datasets, specifically those originating from a consistent donor or protocol. |
| Replication | When benchmarking individual computational methods, we did not report data from multiple runs since these methods exhibit minimal or no variation. Exceptions were made for benchmarking subsampling methods, where we used 9 replicates, and for run time benchmarking, where 3 measurements were taken. All conclusions reported in this study are reproducible across at least 3 independent runs with different random seeds. |
| Randomization | In this study, there was no allocation of participants/samples/organisms into experimental groups. Instead, all computational methods were benchmarked uniformly across all datasets. Consequently, randomization was not a relevant consideration for our experimental design. |
| Blinding | Due to the nature of our study where all computational methods were benchmarked uniformly across all datasets, blinding was not necessary or applicable to our experimental design. |

# Reporting for specific materials, systems and methods

We require information from authors about some types of materials, experimental systems and methods used in many studies. Here, indicate whether each material, system or method listed is relevant to your study. If you are not sure if a list item applies to your research, read the appropriate section before selecting a response.

## Materials & experimental systems

| n/a | Involved in the study |
|-----|----------------------|
| ☒ | Antibodies |
| ☒ | Eukaryotic cell lines |
| ☒ | Palaeontology and archaeology |
| ☒ | Animals and other organisms |
| ☒ | Clinical data |
| ☒ | Dual use research of concern |

## Methods

| n/a | Involved in the study |
|-----|----------------------|
| ☒ | ChIP-seq |
| ☒ | Flow cytometry |
| ☒ | MRI-based neuroimaging |

