## [Peer Review File · Nature Methods]

Peer Review Information

Manuscript Title: SnapATAC2: a fast, scalable and versatile tool for analysis of single-cell omics data

Corresponding author name(s): Bing Ren

Editorial Notes:

Reviewer Comments & Decisions:

Decision Letter, initial version:

10th Aug 2023

Dear Dr Ren,

Please accept our sincere apologies for the long review process. Your Article, "SnapATAC2: a fast, scalable and versatile tool for analysis of single-cell omics data", has now been seen by 3 reviewers. As you will see from their comments below, although the reviewers find your work of considerable potential interest, they have raised a number of concerns. We are interested in the possibility of publishing your paper in Nature Methods, but would like to consider your response to these concerns before we reach a final decision on publication.

We therefore invite you to revise your manuscript to address these concerns. It will be important to follow reviewers' suggestions to add more benchmarking comparisons and show the scalability SnapATAC2.

[REDACTED]

We hope to receive your revised paper within 12 weeks. If you cannot send it within this time, please let us know. In this event, we will still be happy to reconsider your paper at a later date so long as nothing similar has been accepted for publication at Nature Methods or published elsewhere.

OPEN SCIENCE REQUIREMENTS

REPORTING SUMMARY AND EDITORIAL POLICY CHECKLISTS

Please note that these forms are dynamic ‘smart pdfs’ and must therefore be downloaded and completed in Adobe Reader. We will then flatten them for ease of use by the reviewers. If you would like to reference the guidance text as you complete the template, please access these flattened versions at <http://www.nature.com/authors/policies/availability.html>.

DATA AVAILABILITY

All novel DNA and RNA sequencing data, protein sequences, genetic polymorphisms, linked genotype and phenotype data, gene expression data, macromolecular structures, and proteomics data must be deposited in a publicly accessible database, and accession codes and associated hyperlinks must be provided in the “Data Availability” section.

Please include a “Data availability” subsection in the Online Methods. This section should inform readers about the availability of the data used to support the conclusions of your study, including accession

codes to public repositories, references to source data that may be published alongside the paper, unique identifiers such as URLs to data repository entries, or data set DOIs, and any other statement about data availability. At a minimum, you should include the following statement: “The data that support the findings of this study are available from the corresponding author upon request”, describing which data is available upon request and mentioning any restrictions on availability. If DOIs are provided, please include these in the Reference list (authors, title, publisher (repository name), identifier, year). For more guidance on how to write this section please see: <http://www.nature.com/authors/policies/data/data-availability-statements-data-citations.pdf>

CODE AVAILABILITY

Please include a “Code Availability” subsection in the Online Methods which details how your custom code is made available. Only in rare cases (where code is not central to the main conclusions of the paper) is the statement “available upon request” allowed (and reasons should be specified).

ORCID

Nature Methods is committed to improving transparency in authorship. As part of our efforts in this direction, we are now requesting that all authors identified as ‘corresponding author’ on published papers create and link their Open Researcher and Contributor Identifier (ORCID) with their account on the Manuscript Tracking System (MTS), prior to acceptance. This applies to primary research papers only. ORCID helps the scientific community achieve unambiguous attribution of all scholarly contributions. You can create and link your ORCID from the home page of the MTS by clicking on ‘Modify my Springer Nature account’. For more information please visit www.springernature.com/orcid.

Sincerely,
Lei

Lei Tang, Ph.D.
Senior Editor
Nature Methods

Reviewers' Comments:

Reviewer #1:

Remarks to the Author:

In this manuscript, the authors present SnapATAC2, a single-cell genomics data analysis tool for preprocessing and analyzing single-cell data such as single-cell ATAC-seq, single-cell RNA-seq, single-cell HiC, and single-cell methylation. The authors perform a comprehensive benchmarking and performance comparison of SnapATAC2 with existing methods. They demonstrate that SnapATAC2 can yield superior performance to several existing tools and show remarkable scalability and general adaptability at the same time. SnapATAC2 would become a very useful tool to the community. The manuscript is written clearly and represents a high-quality work in the field. I have several minor comments for the authors to consider so that the manuscript can be further improved and be more accessible to readers and method users.

Minor comments:

1. In Figure 5c, the authors compared the clustering performance between snapATAC2 and PCA on scRNA-seq data. PCA is an intermediate data processing step used in pipelines like Seurat, but not a clustering method per se. Methods like Seurat use other clustering algorithms after selecting a few PCs from PCA. The authors should clarify their comparison, i.e., what is the complete procedure of PCA in the benchmarking analysis. Also, what are the parameters used, e.g., how many PCs are selected for clustering analysis, etc. A more detailed description in the Methods section is expected as well.

2. In Figures 2 & 3, it might be better to use the same color to label the same cell type in the schematic in panel A and in the UMAP visualization. It is sometimes confusing to see that one color refers to different cell types in different panels. Similarly, in the comparison Fig. 1c&1d and later figures throughout the manuscript, SnapATAC2 is represented by different colors in different plots. Using a consistent color for each method, at least for SnapATAC2, might help readers follow the results more easily.

3. The online manual of SnapATAC2 needs some improvement to help users.

3a. Many required python dependencies/packages are not mentioned in the installation section, and they are not automatically installed in the “pip install” step. e.g., decorator, scanpy, scvi-tools, etc. The recommended version of dependencies/packages should be specified as well.

3b. A more detailed description of all the core functions and default/recommended parameters, including examples/suggestions about parameter tuning, should be provided. E.g., cutoffs for filter doublets, include more dimension/information in the dimensional reduction, and distance of determine nearest neighbors in the clustering analysis, etc.

3c. Unexpected errors frequently occur when running through the PBMC example datasets (<https://kzhang.org/SnapATAC2/tutorials/pbmc.html>).

i. “Could not find umap or X_umap in .obsm” when running the “sc.pl.umap” function.

ii. “module 'scvi' has no attribute 'model'” when running the “scvi.model.SCVI” function.

3d. The authors should provide examples or tutorials about generating and saving plots into pdf/png format files on the server without graphics.

Reviewer #2:

Remarks to the Author:

In the manuscript, Kai et al. have made substantial advancements with SnapATAC2, an enhanced single-cell analysis package demonstrating superior performance and remarkable scalability compared to the original SnapATAC and other published strategies. Notably, the package employs a matrix-free spectral embedding algorithm for non-linear dimension reduction, validated extensively for its robustness and accuracy. Furthermore, the authors have significantly improved the scalability of the original SnapATAC by implementing key features like on-disk data structures and rewriting critical steps using the Rust language. These modifications enable the processing of large-scale data that surpasses available computer memory. SnapATAC2 is a vital addition to the library of single-cell ATAC-seq analysis tools and could be broadly applied by biologists working on single-cell analysis. Overall, this work is well-suited for publication in Nature Methods, and I do not have major concerns. The following are a couple of suggestions for improving the manuscript and the method's applicability.

1. SnapATAC2 represents a marked improvement compared to the original SnapATAC, yet certain aspects of the new pipeline lack comprehensive descriptions in both the method section and the online tutorial. For instance, while there are detailed documents about the first two steps (Preprocessing/Embedding), the manuscript does not sufficiently describe subsequent steps, including

motif analysis, regulatory network analysis, and visualization methods. As another example, the motif analysis in the tutorial uses a file ("snap.datasets.cis_bp") without explaining its purpose or origin. These areas require further explanation or correction during the revision process.

2. A distinguishing feature of SnapATAC2 is its scalability for large-scale data processing. However, the largest available benchmark scATAC-seq dataset only contains approximately 50,000 nuclei (in Table 1). Therefore, including an extensive comparison or tutorial that processes larger datasets (such as the annotated 1M scATACseq human brain dataset from the same lab) could help demonstrate the method's full capacity. This would also help determine how the subsampling approach affects the accuracy of the embeddings.

3. A major challenge with large-scale scATAC-seq analysis is the increased data sparsity and rarity of cell types. It would be beneficial if the authors could discuss the impact of data sparsity and the rarity of cell types on their algorithm's performance.

Overall, the manuscript is commendable, and the above suggestions are only meant to improve its quality further.

Reviewer #3:

Remarks to the Author:

This manuscript describes a novel approach for analyzing single-cell omics data, introducing a new efficient way of projecting large-scale, high-dimensional, sparse data into low-dimensional space while preserving cellular relationships. The proposed method, SnapATAC2, uses an efficient eigenvector analysis of the graph Laplacian for fast dimensionality reduction of single-cell omics data types, such as ATAC-seq, RNA-seq, Hi-C, and multiomics datasets. SnapATAC2 appears to accurately capture cellular heterogeneity and to demonstrate good performance and scalability. Although the method has some interesting novel elements and promises to be useful there are several areas of concern that need to be addressed.

1. The details of the benchmarking analysis are not adequately described. A complete description would include details about the data, such as which data from the series was selected. If data was filtered, the selection criteria should be clearly stated. Details of the methods and the parameters, in each step should be included. For example, the details of how the clustering done for SnapATAC2 and each of the comparisons are not provided. If variables were transformed before clustering, that would be another detail to include in the methods. Finally, it is essential to report how the gold standards are defined. Complete methods descriptions will allow the benchmarks to be fully assessed. A rigorous assessment of the current manuscript has not been possible.

2. The GSE194122 10x Multiome data set is reported to include 9,876 cells, although the total number of cells in this dataset is an much greater. I did not find any section of the methods describing how this small proportion of cells were selected from the larger dataset. It would be best to not exclude data with explanation, filtering out only low quality cells by some clearly described criteria. The same comment applies to other datasets where subsets of cells might have been selected.

3. In benchmarking it is important to assess the effects of read depth and noise level on performance; this has not been done.

4. The performance of the various methods and their rank differ between datasets. It would be of interest to determine if there are characteristics of the different datasets that account for the differences in methods performance. For example, which methods work best on simple systems or on complex ones with lots of cell types? Does SnapATAC2 do better than the others when the numbers of reads per cell is low or high? Is the number of cells important? Some insight into the results would improve the paper.

5. In developmental systems trajectories between cell types are often observed and investigators are interested in changes in transcription factor activities along these trajectories. In the manuscript the dimension reduction method is evaluated only in terms of cell clusters, although presumably the latent space would also be needed in trajectory analysis. It is therefore important to assess how the SnapATAC2 approach to trajectory inference compares with those implemented in other packages including SCENIC+ and MIRA.

6. Several promising deep learning methods have been left out of the comparisons for this reason: Line 121: “We excluded neural network-based methods from this comparison due to their significantly longer runtime and the requirement for specialized computer hardware.” This statement is not correct: it is not true that neural network methods necessarily take longer times to run than other methods, and it is also not true that all neural network methods require specialized hardware. Furthermore, by specialized hardware, I assume, that the authors are referring to GPUs. GPUs are becoming widely available, on the cloud, in servers and even in laptops. They have been available in MacBook laptops since 2021. Some of the more promising methods for single cell analysis involve neural network models, and they should be included in the comparisons.

7. It is not clear what is meant by “a comprehensive end-to-end analysis”: “Line 132: One of the objectives of SnapATAC2 is to offer comprehensive end-to-end analysis for scATAC-seq data.” The SnapATAC2 analysis starts with a BAM file but surely a mapping step is needed to generate the BAM file. On the other hand, SCENIC+, includes several downstream analyses that seem to go beyond the capabilities of SnapATAC2.

8. As there are many scATAC-seq packages that perform analyses that are not included in SnapATAC2, it is important that SnapATAC2 be interoperable with these software. Does SnapATAC2 provide data formats such as AnnData, or if not how does SnapATAC2 facilitate interoperability with other software?

9. Although the manuscript focuses on the speed of the dimension reduction step it seems that BAM filtering step is taking much of the time, not to mention read mapping, which is not included in the package. This is to say that the impact of implementing a fast dimension reduction step, from the perspective of an 'end-to-end' package needs to be considered in the end-to-end process. Comparison of the single cell pipelines from DNA sequence to clustering would be more meaningful.

10. Although batch correction is part of the software package this aspect of performance is not considered in the manuscript. While some methods take batch into account when dimension reduction is carried out, others correct for batch in a separate postprocessing step. In either case the outcome of the batch correction will be influenced by the dimension reduction step. Several scATAC-seq methods take batch into account in the dimension reduction step itself, including BAVARIA, PeakVI, and MIRA. Besides these there are methods such as Harmony, Scanorama and FastMNN which address batch effects in a postprocessing step. It would be helpful for potential users to see how well batch effects are handled in comparison with these alternative methods. The Lücken et al, 2021 dataset, which was derived with the purpose of evaluating batch effect correction methods would be useful for benchmarking.

11. The user is required to provide the dimension of the latent space as input. What is the effect of choosing different latent space dimensions? A procedure for automating this parameter selection would be useful.

12. The section on multimodal data: "Line 224 However, current methods for analyzing these data often focus on a single modality, underutilizing the full potential of multimodal datasets." The manuscript neglects to mention that several methods do make use of both modalities, including MOFA2, Cobalt, MIRA, MultiVI and Seurat (and others). The paper should make comparisons with these methods.

13. Because single cell ATAC-seq data is sparse, data in enhancer-scale genomics intervals is often imputed. The imputed data is used for several downstream tasks, such as motif and transcription factor binding site identification, as well as gene regulatory analyses. Imputation methods are often part of the dimension reduction process, scATAC-seq imputation methods include cisTopic and SCALE. Does SnapATAC2 provide a way to impute data, and how accurate is this imputation in comparison with other methods? Precision-recall analysis with summary statistics such as the area under the precision recall curve and the F1 score are reasonable ways to measure the performance of methods on this task.

14. Many UMAPs are used in the main figures and in the supplementary figures. It is not always clear what these UMAPs demonstrate, and it would be better to show what the UMAPs are meant to convey in a more clear and quantitative manner.

Author Rebuttal to Initial comments

We are deeply appreciative of the time and efforts of the reviewers in their evaluation of our manuscript. We thank their thoughtful comments and constructive suggestions. We are grateful that the reviewers found SnapATAC2 to be “a very useful tool for the community”, “a vital addition” to the field, and “high quality work”. We are particularly thankful for their constructive comments that helped us further improve the manuscript. In revising our manuscript, we have thoroughly and fully addressed the issues raised by the reviewers by including additional benchmarking analyses, thorough detailing of the methods used in the manuscript, as well as improving the online documentation. These additions have greatly strengthened our work. Below we summarize six major changes made in this revision, followed by a point-by-point response to each of the comments raised by reviewers. Revised texts and paragraphs in the revised manuscript are highlighted in red.

- To bolster the rigor of our benchmark analysis, we have incorporated two additional metrics—beyond ARI and AMI—to evaluate the quality of cell embeddings generated by various methods.
- The Figure 2 has now been updated to examine the effects of sequencing depth, noise level, and cell type rarity on cell embeddings. Across all benchmarks, SnapATAC2 consistently outperforms its competitors.
- We have added benchmark study comparing the scalability of neural network-based models with SnapATAC2. Our results indicate that SnapATAC2 operates at a speed that is orders of magnitude faster than its neural network-based counterparts.
- To showcase the advantages of SnapATAC2’s joint-embedding algorithm, we have conducted comparisons with other methods using two single-cell multi-omics datasets. Our findings reveal that SnapATAC2 excels in both biological conservation and scalability.
- We have also assessed SnapATAC2’s batch correction algorithm against other prevalent methods, confirming that SnapATAC2 ranks among the top-performing algorithms.
- The Methods section has been expanded to offer a thorough overview of the datasets employed, the benchmarking procedures implemented, and the metrics selected for evaluation. Additionally, we have updated the online manual, ensuring that all core functions are accompanied by in-depth descriptions, examples, and explanations for each parameter.

Reviewers' comments:

Reviewer #1:

Remarks to the Author:

In this manuscript, the authors present SnapATAC2, a single-cell genomics data analysis tool for preprocessing and analyzing single-cell data such as single-cell ATAC-seq, single-cell RNA-seq, single-cell HiC, and single-cell methylation. The authors perform a comprehensive benchmarking and performance comparison of SnapATAC2 with existing methods. They demonstrate that SnapATAC2 can yield superior performance to several existing tools and show remarkable scalability and general adaptability at the same time. SnapATAC2 would become a very useful tool to the community. The manuscript is written clearly and represents a high-quality work in the field. I have several minor comments for the authors to consider so that the manuscript can be further improved and be more accessible to readers and method users.

We are appreciative of the reviewer's positive evaluation of our manuscript and the recognition of SnapATAC2's potential contributions to the field of single-cell genomics.

Minor comments:

1. In Figure 5c, the authors compared the clustering performance between snapATAC2 and PCA on scRNA-seq data. PCA is an intermediate data processing step used in pipelines like Seurat, but not a clustering method per se. Methods like Seurat use other clustering algorithms after selecting a few PCs from PCA. The authors should clarify their comparison, i.e., what is the complete procedure of PCA in the benchmarking analysis. Also, what are the parameters used, e.g., how many PCs are selected for clustering analysis, etc. A more detailed description in the Methods section is expected as well.

We thank the reviewer for this comment. We now have added detailed description of benchmarking analysis in the Methods section (highlighted in red font), including in-depth overview of the datasets utilized, the benchmarking procedures employed, and the metrics used for the benchmark.

To assess various dimensionality reduction techniques, such as PCA, we have further incorporated four distinct metrics into our analysis: Adjusted Rand Index (ARI), Adjusted Mutual Information (AMI), Cell Type Average Silhouette Width (Cell Type ASW), and Graph Integration Local Inverse Simpson's Index (Graph iLISI). The Methods section elaborates on these metrics. Briefly, ARI and AMI measure the extent of similarity between two sets of

clusters, accounting for both overlaps and mismatches. As the reviewer astutely noted, PCA and similar dimensionality reduction methods do not inherently produce clusters. To address this, we applied the Leiden clustering algorithm to the resulting cell embeddings to generate clusters. These clusters were then compared against ground-truth cell labels to compute the ARI or AMI scores. On the other hand, the remaining metrics -- Cell Type ASW and Graph iLISI - - are not reliant on cluster comparisons and can be directly calculated from the cell embeddings.

Regarding the number of components used in the benchmarking analysis, we generally opted for a dimensionality of 30, which efficiently captures most data variance. Certain methods, like cisTopic, PCA, and LSI, necessitate fine-tuning of the component count or dimensionality. For these methods, we adhered to guidelines set forth in their respective publications to determine optimal settings. Specifically for PCA, we employed the commonly used "elbow" method to select the appropriate number of components. This technique involves plotting the cumulative explained variance against the number of components selected. The "elbow" point, where the increase in explained variance starts to level off, serves as a heuristic for choosing an optimal number of principal components, balancing between dimensionality reduction and variance explanation.

All of these details, along with other relevant information, have been incorporated into the revised version of our manuscript.

2. In Figures 2 & 3, it might be better to use the same color to label the same cell type in the schematic in panel A and in the UMAP visualization. It is sometimes confusing to see that one color refers to different cell types in different panels. Similarly, in the comparison Fig. 1c&1d and later figures throughout the manuscript, SnapATAC2 is represented by different colors in different plots. Using a consistent color for each method, at least for SnapATAC2, might help readers follow the results more easily.

We thank the reviewer for this suggestion. We have changed the color labeling scheme accordingly and made sure they are consistent throughout the manuscript.

3. The online manual of SnapATAC2 needs some improvement to help users.

We thank the reviewer for this comment. We have now thoroughly revised the online manual to ensure its clarity, correctness, and completeness. Please see detailed response below.

3a. Many required python dependencies/packages are not mentioned in the installation section, and they are not automatically installed in the "pip install" step. e.g., decorator, scanpy, scvi-tools, etc. The recommended version of dependencies/packages should be specified as well.

Thank you for pointing out the oversight in our installation documentation. We have updated the section to include comprehensive information on how to install required and optional Python dependencies/packages. You can find this updated information on our installation webpage: <https://kzhang.org/SnapATAC2/install.html#optional-dependencies>.

To make the installation process smoother, we have introduced three new pip keywords: "extra", "recommend", and "all":

- To install only the essential dependencies, you can use the command “pip install snapatac2”.
- If you wish to install optional packages that enhance functionality but are not strictly necessary, use “pip install snapatac2[extra]”.
- For downstream analysis that may require additional but non-mandatory packages like scanpy and scvi-tools, you can use “pip install snapatac2[recommend]”.
- To install all optional dependencies in one go, the command “pip install snapatac2[all]” is available.

We hope this clarified and streamlined installation process makes it easier for users to set up the environment tailored to their needs.

3b. A more detailed description of all the core functions and default/recommended parameters, including examples/suggestions about parameter tuning, should be provided. E.g., cutoffs for filter doublets, include more dimension/information in the dimensional reduction, and distance of determine nearest neighbors in the clustering analysis, etc.

We thank the review for this feedback. To address this, we have updated our online documentation to include in-depth explanations of each core function. We also provide a recommended set of default parameters alongside guidance on how to tune these parameters for specific use-cases, as exemplified below:

```

snapatac2.pp.import_data(fragment_file, *, file=None,
genome=None, gene_anno=None, chrom_size=None,
min_num_fragments=200, min_tsse=1, sorted_by_barcode=True,
low_memory=True, whitelist=None, shift_left=0, shift_right=0,
chunk_size=2000, tempdir=None, backend='hdf5', n_jobs=8)
    Import dataset and compute QC metrics. [source]

    This function will store fragments as base-resolution TN5 insertions in the
    resulting h5ad file (in .obsm['insertion']), along with the chromosome sizes
    (in .uns['reference_sequences']). Various QC metrics, including TSSe,
    number of unique fragments, duplication rate, fraction of mitochondrial DNA
    reads, will be computed. The .obsm['insertion'] matrix created in this step is
    essential for downstream analysis, such as tile matrix generation and peak
    calling.
    
```

Argument names and their default parameters

Function description

```

Parameters:
• fragment_file ( Path | List [ Path ] ) – File name of the fragment file.
  This can be a single file or a list of files.
• file ( Union [ Path , List [ Path ] , None ] ) – File name of the output
  h5ad file used to store the result. If provided, result will be saved to a
  backed AnnData, otherwise an in-memory AnnData is used. If
  fragment_file is a list of files, file must also be a list of files if
  provided.
• genome ( Optional [ Genome ] ) – A Genome object, providing gene
  annotation and chromosome sizes. If not set, gff_file and
  chrom_size must be provided. genome has lower priority than
  gff_file and chrom_size.
• gene_anno ( Optional [ Path ] ) – File name of the gene annotation file
  in GFF or GTF format. This is required if genome is not set. Setting
    
```

Explanation of the parameters

```

Examples

>>> import snapatac2 as snap
>>> data = snap.pp.import_data(snap.datasets.pbmc500(), genome=snap
>>> print(data)
AnnData object with n_obs × n_vars = 816 × 0
  obs: 'tsse', 'n_fragment', 'frac_dup', 'frac_mito'
  uns: 'reference_sequences'
  obsm: 'insertion'
    
```

Examples

3c. Unexpected errors frequently occur when running through the PBMC example datasets (<https://kzhang.org/SnapATAC2/tutorials/pbmc.html>).

- i. “Could not find umap or X_umap in .obsm” when running the “sc.pl.umap” function.
- ii. “module 'scvi' has no attribute 'model'” when running the “scvi.model.SCVI” function.

We sincerely apologize for the inconvenience you've experienced while working through the PBMC example datasets. Thank you for bringing these issues to our attention. We have taken immediate steps to resolve the problems you've mentioned and have updated the tutorial accordingly. Here are the specific actions we have taken:

- "Could not find umap or X_umap in .obsm" when running the "sc.pl.umap" function: This issue usually arises when the UMAP data has not been pre-computed and stored in

the “.obsm” attribute. We've clarified the steps in the tutorial where this should be done. Make sure to follow the UMAP calculation steps before calling the “sc.pl.umap” function.

- "module 'scvi' has no attribute 'model' when running the 'scvi.model.SCVI' function": this is likely due to a version mismatch or the installation of the outdated "SCVI" package instead of the current "scvi-tools" package. To address this, we've updated our installation instructions to specify the recommended version of "scvi-tools" that is compatible with our tutorial. Once you update to the version we've recommended, you should be able to run the “scvi.model.SCVI” function without issues.

The revised tutorial is here: <https://kzhang.org/SnapATAC2/tutorials/annotation.html>.

3d. The authors should provide examples or tutorials about generating and saving plots into pdf/png format files on the server without graphics.

Thank you for this suggestion. We have updated our documentation to include a new section that guides users through the process of creating and saving plots in both PDF and PNG formats without requiring a graphical interface. You can find this additional content in our updated tutorials: <https://kzhang.org/SnapATAC2/tutorials/pbmc.html#Clustering-analysis>.

Reviewer #2:

Remarks to the Author:

In the manuscript, Kai et al. have made substantial advancements with SnapATAC2, an enhanced single-cell analysis package demonstrating superior performance and remarkable scalability compared to the original SnapATAC and other published strategies. Notably, the package employs a matrix-free spectral embedding algorithm for non-linear dimension reduction, validated extensively for its robustness and accuracy. Furthermore, the authors have significantly improved the scalability of the original SnapATAC by implementing key features like on-disk data structures and rewriting critical steps using the Rust language. These modifications enable the processing of large-scale data that surpasses available computer memory. SnapATAC2 is a vital addition to the library of single-cell ATAC-seq analysis tools and could be broadly applied by biologists working on single-cell analysis. Overall, this work is well-suited for publication in Nature Methods, and I do not have major concerns. The following are a couple of suggestions for improving the manuscript and the method's applicability.

We are grateful for your positive assessment of our work and the advancements made with SnapATAC2.

1. SnapATAC2 represents a marked improvement compared to the original SnapATAC, yet certain aspects of the new pipeline lack comprehensive descriptions in both the method section and the online tutorial. For instance, while there are detailed documents about the first two steps (Preprocessing/Embedding), the manuscript does not sufficiently describe subsequent steps, including motif analysis, regulatory network analysis, and visualization methods. As another example, the motif analysis in the tutorial uses a file ("snap.datasets.cis_bp") without explaining its purpose or origin. These areas require further explanation or correction during the revision process.

We are grateful for the reviewer's comments and suggestions! We've expanded the manuscript to include thorough descriptions of each component within the SnapATAC2 package, as well as providing more details in the Methods section, that covers all aspects of data processing, such as data format, batch correction, doublet detection, peak calling, differential accessibility analysis, motif enrichment analysis, and regulatory network analysis. Furthermore, we have thoroughly updated the online manual, ensuring that all core functions are accompanied by detailed descriptions, examples, and parameter explanations.

2. A distinguishing feature of SnapATAC2 is its scalability for large-scale data processing. However, the largest available benchmark scATAC-seq dataset only contains approximately 50,000 nuclei (in Table 1). Therefore, including an extensive comparison or tutorial that processes larger datasets (such as the annotated 1M scATACseq human brain dataset from the same lab) could help demonstrate the method's full capacity. This would also help determine how the subsampling approach affects the accuracy of the embeddings.

We appreciate the reviewer's suggestion to showcase SnapATAC2's capability for large-scale data processing. In Figure 1e, we have compared SnapATAC2's scalability with that of ArchR by analyzing a substantial dataset of over 650,000 cells. Furthermore, as suggested by the reviewer, we have updated our online documentation to feature a tutorial that processes the same dataset used in Figure 1e. This dataset, part of our previously published human single-cell chromatin atlas, contains more than 650,000 cells and effectively showcases SnapATAC2's scalability.

The new tutorial is here: <https://kzhang.org/SnapATAC2/tutorials/atlas.html>.

3. A major challenge with large-scale scATAC-seq analysis is the increased data sparsity and rarity of cell types. It would be beneficial if the authors could discuss the impact of data sparsity and the rarity of cell types on their algorithm's performance.

We are grateful for the reviewer's constructive comment. In response, we have expanded our analysis to include benchmarks using synthetic datasets featuring different noise levels, sequencing coverages, and varying abundances of cell types. These updated findings are now presented in Figure 2 and Figure S1 of the revised manuscript.

In Figure 2, we performed the benchmarking using synthetic scATAC-seq data, consisting of eight simulated datasets with varying sequencing depths and noise levels, along with ground-truth cell type labels. Our findings, illustrated in Figure 2b, reveal that SnapATAC2 consistently outperformed other methods across varying sequencing depths, achieving the highest ARI scores. For example, at a sequencing depth of 5,000 reads per cell, all tested algorithms accurately identified the six cell types, garnering ARI scores between 0.94 and 1.00. However, when the sequencing depth was reduced to 1,000 reads per cell, only SnapATAC2 and Signac maintained an ARI score above 0.9. Particularly, PeakVI was highly sensitive to sequencing depth, its ARI score plummeting to 0.006 at 250 reads per cell (Figure 2b,c). In contrast, SnapATAC2 maintained a score of 0.47. We observed similar robustness in performance when assessing noise levels. SnapATAC2 achieved perfect ARI scores (1.0) at all examined noise levels, followed closely by Signac and the original SnapATAC (Figure 2d). In comparison, SCALE and PeakVI showed the most sensitivity to noise, with their ARI scores dropping to 0.57 and 0.46, respectively, at a noise level of 0.4 (Figure 2d,e).

To assess SnapATAC2's ability to identify rare cell types, we generated synthetic datasets featuring varying proportions of CD8+ T cells, from as low as 0.5% to as high as 15% of the total cell population. We chose CD8+ T cells for this analysis because of their chromatin accessibility profiles' similarity to CD4+ T cells, which makes distinguishing between the two a significant challenge. We then applied various dimensionality reduction algorithms to these modified datasets. The effectiveness of each method was assessed by calculating the average silhouette width for the CD8+ T cell population based on their embeddings. Specifically, the silhouette width metric was used to quantify the separation between the CD8+ T cells and their closest neighboring cell population, thereby providing an insight into each algorithm's capacity to differentiate this rare cell type effectively. Figure S1 reveals that SnapATAC2 was particularly effective at identifying this rare cell type, even when they constituted only 0.5% of the total cells.

In conclusion, the added analyses underscore SnapATAC2's robust performance across different sequencing depths, noise levels, and variable cell-type abundances, confirming its capability to produce consistently high-quality embeddings.

Overall, the manuscript is commendable, and the above suggestions are only meant to improve its quality further.

Thank you for taking the time to provide these constructive feedbacks!

Reviewer #3:

Remarks to the Author:

This manuscript describes a novel approach for analyzing single-cell omics data, introducing a new efficient way of projecting large-scale, high-dimensional, sparse data into low-dimensional space while preserving cellular relationships. The proposed method, SnapATAC2, uses an efficient eigenvector analysis of the graph Laplacian for fast dimensionality reduction of single-cell omics data types, such as ATAC-seq, RNA-seq, Hi-C, and multiomics datasets. SnapATAC2 appears to accurately capture cellular heterogeneity and to demonstrate good performance and scalability. Although the method has some interesting novel elements and promises to be useful, there are several areas of concern that need to be addressed.

We thank the reviewer for the constructive comments and for recognizing the novelty and potential utility of our SnapATAC2 method. Below, we provide a point-to-point response to address each concern raised.

1. The details of the benchmarking analysis are not adequately described. A complete description would include details about the data, such as which data from the series was selected. If data was filtered, the selection criteria should be clearly stated. Details of the methods and the parameters, in each step should be included. For example, the details of how the clustering done for SnapATAC2 and each of the comparisons are not provided. If variables were transformed before clustering, that would be another detail to include in the methods. Finally, it is essential to report how the gold standards are defined. Complete methods descriptions will allow the benchmarks to be fully assessed. A rigorous assessment of the current manuscript has not been possible.

We thank reviewer for this comment. We've greatly expanded the Methods section with a thorough description of the benchmarking analysis as suggested by this reviewer. This now includes detailed information on the datasets used, the specific procedures followed for benchmarking, and the metrics applied for evaluation.

In the revised manuscript, we clarify that we used published cell type annotations as ground truth labels for all datasets except for synthetic ones, where labels are inherently known. We did not filter out any annotated cells except in the case of the GSE194122 and Zemke_human datasets. Due to the multiple donors or protocols involved in these datasets, they exhibited pronounced batch effects. To minimize bias, we used only subsets of cells from these two datasets that originated from a donor. Note that the complete versions of these datasets were used in our evaluation of batch effect correction methods.

To enhance the rigor of our benchmarking process, we've added two new metrics: Cell Type Average Silhouette Width (Cell Type ASW) and Graph Integration Local Inverse Simpson's

Index (Graph iLISI), supplementing our use of the ARI and AMI metrics. Each metric's calculation methodology is described in the Methods section. To compute ARI and AMI, we employed a standardized clustering approach across all methods. We first generated a k-NN graph using the cell embeddings, setting k at 50. Using this graph, we applied the Leiden algorithm to create cell clusters, and fine-tuned the Leiden algorithm's resolution parameter between 0.1 and 3.0 in 0.1 increments to align with the known number of cell types in the benchmarking datasets. It's important to note that the newly incorporated metrics, Cell Type ASW and Graph iLISI, are calculated directly from the cell embeddings and do not require cluster comparisons.

2. The GSE194122 10x Multiome data set is reported to include 9,876 cells, although the total number of cells in this dataset is much greater. I did not find any section of the methods describing how this small proportion of cells were selected from the larger dataset. It would be best to not exclude data with explanation, filtering out only low quality cells by some clearly described criteria. The same comment applies to other datasets where subsets of cells might have been selected.

We appreciate the reviewer's suggestion. In the updated manuscript, we've clarified that we included all annotated cells in our analysis, with the exception of the GSE194122 and Zemke_human datasets. These particular datasets contained data from multiple donors or protocols, introducing significant batch effects. To mitigate this bias, we restricted our analysis to a subset of cells from these datasets that came from a uniform donor or protocol. Importantly, the full versions of these two datasets were still utilized in our assessment of batch effect correction techniques.

3. In benchmarking it is important to assess the effects of read depth and noise level on performance; this has not been done.

We are grateful for the reviewer's constructive comment. In response, we have expanded our analysis to include benchmarks using synthetic datasets featuring different noise levels, sequencing coverages, and varying abundances of cell types. These updated findings are now presented in Figure 2 and Figure S1 of the revised manuscript.

In Figure 2, we performed the benchmarking using synthetic scATAC-seq data, consisting of eight simulated datasets with varying sequencing depths and noise levels, along with ground-truth cell type labels. Our findings, illustrated in Figure 2b, reveal that SnapATAC2 consistently outperformed other methods across varying sequencing depths, achieving the highest ARI scores. For example, at a sequencing depth of 5,000 reads per cell, all tested algorithms accurately identified the six cell types, garnering ARI scores between 0.94 and 1.00. However, when the sequencing depth was reduced to 1,000 reads per cell, only SnapATAC2 and Signac maintained an ARI score above 0.9. Particularly, PeakVI was highly sensitive to sequencing depth, its ARI score plummeting to 0.006 at 250 reads per cell (Figure 2b,c). In contrast,

SnapATAC2 maintained a score of 0.47. We observed similar robustness in performance when assessing noise levels. SnapATAC2 achieved perfect ARI scores (1.0) at all examined noise levels, followed closely by Signac and the original SnapATAC (Figure 2d). In comparison, SCALE and PeakVI showed the most sensitivity to noise, with their ARI scores dropping to 0.57 and 0.46, respectively, at a noise level of 0.4 (Figure 2d,e).

To assess SnapATAC2's ability to identify rare cell types, we generated synthetic datasets featuring varying proportions of CD8+ T cells, from as low as 0.5% to as high as 15% of the total cell population. We chose CD8+ T cells for this analysis because of their chromatin accessibility profiles' similarity to CD4+ T cells, which makes distinguishing between the two a significant challenge. We then applied various dimensionality reduction algorithms to these modified datasets. The effectiveness of each method was assessed by calculating the average silhouette width for the CD8+ T cell population based on their embeddings. Specifically, the silhouette width metric was used to quantify the separation between the CD8+ T cells and their closest neighboring cell population, thereby providing an insight into each algorithm's capacity to differentiate this rare cell type effectively. Figure S1 reveals that SnapATAC2 was particularly effective at identifying this rare cell type, even when they constituted only 0.5% of the total cells.

In conclusion, the added analyses underscore SnapATAC2's robust performance across different sequencing depths, noise levels, and variable cell-type abundances, confirming its capability to produce consistently high-quality embeddings.

4. The performance of the various methods and their rank differ between datasets. It would be of interest to determine if there are characteristics of the different datasets that account for the differences in methods performance. For example, which methods work best on simple systems or on complex ones with lots of cell types? Does SnapATAC2 do better than the others when the numbers of reads per cell is low or high? Is the number of cells important? Some insight into the results would improve the paper.

We appreciate the reviewer's insightful comment on examining the relationship between dataset characteristics and method performance. To address this, we have summarized the Bio-conservation score, which is the average of four evaluation metrics, for each method across ten benchmark datasets. Additionally, we have detailed key features of each dataset: the number of cells, the number of cell types, the average number of reads per cell, and the number of features.

Upon examination, we did not observe a strong correlation between these dataset characteristics and the performance rankings of the various methods. This suggests that the variation in performance is likely due to other factors, such as the intricate nature of the datasets, the varying quality of cell labels, and the inherent variability of the algorithms (especially neural network-based algorithms that often converge to local optima).

Here is the data presented in a table format for clarity:

Dataset	No. of cells	No. of cell types	Reads per cell	No. of features	Bio-conservation score (average of 4 metrics)								
					SnapATAC2	ArchR (LSI)	PeakV	SCALE	Signac (LSI)	SnapATAC	cisTopic (LDA)	epiScanpy (PCA)	scBasset
10x_Brain5k	2317	10	38282	155093	0.997	0.835	0.738	0.711	0.435	0.000	0.813	0.536	0.833
10x_PBMC10k	9631	19	20479	107194	1.000	0.834	0.763	0.703	0.460	0.253	0.704	0.017	0.172
Buenrostro_2018	2034	10	15409	237440	0.892	0.389	0.849	0.728	0.139	0.025	0.865	0.054	0.796
Chen_NBT_2019	9190	22	2641	241757	0.907	0.405	0.514	0.126	0.247	0.158	0.465	0.000	1.000
GSE194122	9876	19	8260	116490	0.907	0.474	0.929	0.701	0.171	0.067	0.851	0.009	0.908
Ma_Cell_2020	32231	22	4152	340341	1.000	0.764	0.893	0.648	0.544	0.744	0.842	0.740	0.000
Trevino_Cell_2021	8981	13	16519	467315	0.904	0.463	0.640	0.288	0.055	0.447	0.582	0.481	0.985
Yao_Nature_2021	54844	11	3026	148814	0.998	0.590	0.789	0.576	0.635	0.298	0.822	0.030	0.958
Zemke_2023_human	15284	20	16854	380517	0.908	0.635	0.994	0.447	0.005	0.443	0.872	0.134	0.261
Zemke_2023_mouse	45089	19	28880	330448	0.790	0.518	0.996	0.125	0.473	0.439	0.930	0.250	0.751

5. In developmental systems trajectories between cell types are often observed and investigators are interested in changes in transcription factor activities along these trajectories. In the manuscript the dimension reduction method is evaluated only in terms of cell clusters, although presumably the latent space would also be needed in trajectory analysis. It is therefore important to assess how the SnapATAC2 approach to trajectory inference compares with those implemented in other packages including SCENIC+ and MIRA.

We are grateful to the reviewer for the insightful comment and recognize the importance of trajectory inference in single-cell data analysis. Indeed, the Laplacian matrix used in SnapATAC2's spectral embedding has interesting parallels to the diffusion matrix commonly used in trajectory analysis¹. While we acknowledge that evaluating the applicability of SnapATAC2's spectral embedding for trajectory inference could offer valuable insights, we consider such an in-depth analysis to be beyond the scope of our current study.

To address the concern regarding our focus solely on cell clustering metrics, we have expanded our evaluation criteria to include two additional metrics that do not rely on cluster comparisons. Specifically, we have included Cell Type Average Silhouette Width (Cell Type ASW) and Graph Integration Local Inverse Simpson's Index (Graph iLISI) in our benchmarking. These metrics serve to quantify the quality of the embeddings in capturing meaningful biological variability, which could indirectly shed light on their applicability for trajectory inference tasks. We believe that these enhancements add another layer of rigor to our study and provide more comprehensive insights into SnapATAC2's capabilities.

6. Several promising deep learning methods have been left out of the comparisons for this reason: Line 121: "We excluded neural network-based methods from this comparison due to their

significantly longer runtime and the requirement for specialized computer hardware.” This statement is not correct: it is not true that neural network methods necessarily take longer times to run than other methods, and it is also not true that all neural network methods require specialized hardware. Furthermore, by specialized hardware, I assume, that the authors are referring to GPUs. GPUs are becoming widely available, on the cloud, in servers and even in laptops. They have been available in MacBook laptops since 2021. Some of the more promising methods for single cell analysis involve neural network models, and they should be included in the comparisons.

We appreciate the reviewer's recommendation to include neural network-based methods in our scalability benchmark. To address this, we've updated Figure 1c to incorporate a comparison with three neural network-based techniques: PeakVI, scBasset, and SCALE. We utilized an A100 GPU to expedite computations and monitored runtimes over 50 epochs, a generally accepted baseline for algorithmic convergence. It's worth noting that the memory requirements for these neural network-based methods scale with feature count. When the number of features exceeded 500,000, we encountered GPU memory limitations even on our A100 GPU, which has 40GB of memory. As a result, we limited the feature set to 500,000 for these methods, while SnapATAC2 and others used the full feature set.

As shown in Figure 1c, SnapATAC2, along with ArchR, Signac, and EpiScanpy, exhibited minimal increases in runtime even as the dataset size grew. Conversely, neural network-based methods, although linearly scalable, took significantly longer to process the same data. For instance, SnapATAC2 required only 13.4 minutes to analyze a 200,000-cell dataset, compared to about four hours for PeakVI. Importantly, the reported runtimes for neural network-based methods do not include data preprocessing time, and actual training may require more than 50 epochs, potentially extending the computational time further.

Lastly, we'd like to highlight that while GPUs are increasingly common, high-end GPUs like the A100 remain less accessible for many research labs. Additionally, as demonstrated in our benchmarks, even these high-end GPUs can face memory constraints, particularly when analyzing epigenomics datasets with an extremely large number of features.

7. It is not clear what is meant by “a comprehensive end-to-end analysis”: “Line 132: One of the objectives of SnapATAC2 is to offer comprehensive end-to-end analysis for scATAC-seq data.” The SnapATAC2 analysis starts with a BAM file but surely a mapping step is needed to generate the BAM file. On the other hand, SCENIC+, includes several downstream analyses that seem to go beyond the capabilities of SnapATAC2.

We appreciate the reviewer's comment on the phrase "a comprehensive end-to-end analysis" and acknowledge that it may not accurately represent the capabilities of SnapATAC2. We have removed this statement from the revised manuscript. The updated text now reads:

“One of the aims of SnapATAC2 is to offer a wide-ranging analysis for scATAC-seq data, covering multiple stages of the process. ArchR has been previously cited as one of the most scalable and comprehensive software packages for similar tasks...”

8. As there are many scATAC-seq packages that perform analyses that are not included in SnapATAC2, it is important that SnapATAC2 be interoperable with these software. Does SnapATAC2 provide data formats such as AnnData, or if not how does SnapATAC2 facilitate interoperability with other software?

We appreciate the reviewer's comment and have updated the Methods section to clarify our use of AnnData as the data format for SnapATAC2. This allows for seamless integration with SCANPY, scvi-tools, and other packages that utilize the AnnData format.

Importantly, SnapATAC2 introduces an enhanced out-of-core AnnData object that addresses a long-standing issue in the "anndata" package, as highlighted here: <https://github.com/scverse/anndata/issues/690>. The out-of-core AnnData object in SnapATAC2 comes with several key features:

1. Full HDF5 backing: The AnnData object is completely backed by its HDF5 file, ensuring real-time disk synchronization for any changes made in memory.
2. Lazy Loading: This feature minimizes memory overhead, making it possible to open large files with negligible memory usage.
3. Optional In-memory Cache: For enhanced performance, an optional in-memory cache can be used to expedite repeated data access.
4. AnnDataSet Object: We've also introduced an AnnDataSet object designed to concatenate multiple AnnData objects lazily, thereby increasing both efficiency and functionality.

These enhancements make SnapATAC2 a more robust and flexible tool for single-cell genomics analysis.

9. Although the manuscript focuses on the speed of the dimension reduction step it seems that BAM filtering step is taking much of the time, not to mention read mapping, which is not included in the package. This is to say that the impact of implementing a fast dimension reduction step, from the perspective of an 'end-to-end' package, needs to be considered in the end-to-end process. Comparison of the single cell pipelines from DNA sequence to clustering would be more meaningful.

We appreciate your feedback and apologize for any confusion caused by our use of the term "end-to-end" to describe SnapATAC2. We've revised the manuscript to eliminate this statement for clarity.

We chose not to include the read mapping step in our benchmarking because most pipelines employ highly similar strategies at this stage, offering little room for variation. Hence, we felt that benchmarking this step was not necessary. While it's true that both read mapping and BAM filtering can be time-consuming, it's important to note that their impact on the overall runtime should not overshadow the significance of the dimensionality reduction step. Unlike read mapping and BAM filtering, which are generally performed only once, dimensionality reduction might be executed multiple times during the analysis. This could occur for various reasons, such as parameter tuning, different feature selection strategies, cell filtering methods, or the need for sub-clustering. As a result, the cumulative time spent on dimensionality reduction can actually exceed that of read mapping and BAM filtering, making its efficiency a crucial aspect of the process.

10. Although batch correction is part of the software package, this aspect of performance is not considered in the manuscript. While some methods take batch into account when dimension reduction is carried out, others correct for batch in a separate postprocessing step. In either case the outcome of the batch correction will be influenced by the dimension reduction step. Several scATAC-seq methods take batch into account in the dimension reduction step itself, including BAVARIA, PeakVI, and MIRA. Besides these there are methods such as Harmony, Scanorama and FastMNN which address batch effects in a postprocessing step. It would be helpful for potential users to see how well batch effects are handled in comparison with these alternative methods. The Lücken et al, 2021 dataset, which was derived with the purpose of evaluating batch effect correction methods would be useful for benchmarking.

We appreciate the reviewer's insightful comment on the importance of batch correction in the analysis of single-cell data. To address this, we have included new benchmarks focused on batch correction in Figure S13 of our revised manuscript. For the scRNA-seq data, we utilized the Lücken et al., 2021 dataset as suggested by the reviewer. We generated cell embeddings using SnapATAC2 and then applied various post-processing batch correction algorithms such as MNNCorrect, Harmony, and Scanorama. Our findings indicate that SnapATAC2, when used in conjunction with our modified version of the MNNCorrect algorithm, ranks third, closely following scVI and FastMNN.

For the scATAC-seq analysis, we utilized the full datasets from GSE194122 and Zemke_human to conduct benchmarking tests. Unlike in our dimensionality reduction benchmarks where only a subset of the data was used, here we included data from multiple donors and various protocols. Out of the 20 methods evaluated, SnapATAC2 when paired with MNNCorrect, ranked second, almost matching the performance of the top performer, PeakVI.

We believe these additional benchmarks will offer users valuable insights into how well SnapATAC2 handles batch effects in comparison to other state-of-the-art methods. Thank you for your constructive feedback, which has helped us improve the comprehensiveness of our study.

11. The user is required to provide the dimension of the latent space as input. What is the effect of choosing different latent space dimensions? A procedure for automating this parameter selection would be useful.

SnapATAC2 provides an automated approach to eigenvector reweighting, which thereby eliminates the need for manual dimension selection. This approach is detailed in the Methods section, copied below:

“Not all eigenvectors produced by spectral embedding are informative and relevant for clustering tasks. Selecting appropriate eigenvectors is essential, as using uninformative or irrelevant ones can lead to suboptimal clustering results. We found that the widely-used elbow method for determining the number of eigenvectors is not consistently reliable in practice. To identify relevant eigenvectors, we propose a simple heuristic based on the eigenvalues of the graph Laplacian matrix. In this approach, each eigenvector is weighted by the square root of its corresponding eigenvalue, and these weighted eigenvectors are then employed for further analyses.”

This feature renders SnapATAC2 largely insensitive to the choice of dimensionality, provided it is sufficient to capture the inherent complexity of the data. Typically, a dimensionality of 30 is adequate, a parameter we consistently use in our benchmarks. This approach aligns with neural network-based algorithms where adjusting the latent space's dimensionality is often not required.

12. The section on multimodal data: “Line 224 However, current methods for analyzing these data often focus on a single modality, underutilizing the full potential of multimodal datasets.” The manuscript neglects to mention that several methods that do make use of both modalities, including MOFA2, Cobolt, MIRA, MultiVI and Seurat (and others). The paper should make comparisons with these methods.

We appreciate the reviewer's feedback and have taken steps to amend the manuscript by removing the statement in question. To provide a more comprehensive analysis, we have included new benchmarking data comparing the co-embedding outputs from multimodal datasets across several methods, including SnapATAC2, MOFA+ (MOFA2), Cobolt, and MIRA. We opted not to include Seurat and MultiVI in this comparison; Seurat does not generate cell embeddings as output, and we encountered persistent errors with MultiVI due to a known bug in the package (see GitHub issue: <https://github.com/scverse/scvi-tools/issues/1291>).

The outcomes of this additional analysis have been integrated into Figure 5c,d and the manuscript has been updated accordingly to reflect these new findings, copied below:

“In our pursuit to further validate the performance of SnapATAC2, we compared it against other joint embedding techniques, specifically MIRA, Cobolt, and MOFA+. Using the same 10x Genomics multiome dataset, SnapATAC2 consistently ranked highest in bio-conservation scores across all four evaluation metrics (Fig. 5c). To broaden the scope of our comparative analysis, we also incorporated a dataset profiling H3K27me3 occupancy and gene expression in 10,180 cells from the mouse frontal cortex. Once again, SnapATAC2 emerged as the top-performing method, achieving the highest average bio-conservation score (Fig. 5d). Beyond its exceptional accuracy, SnapATAC2 also showcased unparalleled scalability. Across both datasets, it drastically outperformed competitors like MIRA, Cobolt, and MOFA+ in computational speed and memory efficiency, running more than 30 times faster than the next best method (Fig. 5c,d). In summary, these results validate SnapATAC2’s excellence not only in bio-conservation quality but also in computational efficiency, making it a highly robust and scalable solution for analyzing complex single-cell multiomics data.”

13. Because single cell ATAC-seq data is sparse, data in enhancer-scale genomics intervals is often imputed. The imputed data is used for several downstream tasks, such as motif and transcription factor binding site identification, as well as gene regulatory analyses. Imputation methods are often part of the dimension reduction process, scATAC-seq imputation methods include cisTopic and SCALE. Does SnapATAC2 provide a way to impute data, and how accurate is this imputation in comparison with other methods? Precision-recall analysis with summary statistics such as the area under the precision recall curve and the F1 score are reasonable ways to measure the performance of methods on this task.

Thank you for bringing up the important issue of data imputation in single-cell ATAC-seq analyses! While SnapATAC2 offers various preprocessing and advanced analytical tools, it does not have a built-in imputation module. Instead, it is designed for seamless integration with specialized packages within the Python scverse ecosystem. Thanks to its compatibility with the AnnData format, SnapATAC2 can easily work in conjunction with packages like SCANPY and scvi-tools, which do offer imputation methods. For example, in our online tutorial, we show how to transfer data from SnapATAC2 to SCANPY for imputation using the MAGIC algorithm. This approach allows users to take advantage of the best tools for each specific task in their analysis pipeline. We have expanded on this aspect in the Discussion section of the revised manuscript, stating:

“SnapATAC2 offers a unique advantage in its seamless interoperability with other software tools widely used in the single-cell analytics ecosystem. By adopting the AnnData format, it facilitates effortless integration with established packages like SCANPY, scvi-tools, and SCENIC+. This feature is especially advantageous for researchers seeking to carry out

specialized analyses, such as data imputation or trajectory inference, thereby enhancing the core functions of SnapATAC2.”

14. Many UMAPs are used in the main figures and in the supplementary figures. It is not always clear what these UMAPs demonstrate, and it would be better to show what the UMAPs are meant to convey in a more clear and quantitative manner.

We appreciate the reviewer's feedback on the clarity of the UMAP figures in our manuscript. In response, we have streamlined the presentation by removing most of the UMAP figures, retaining only those that make a specific, clear point. To provide a more quantitative assessment, we have replaced the omitted UMAPs with tables that present quantitative differences between various embeddings. This approach should offer a clearer and more direct way to understand the data and compare the results.

Reference

1. Bergen, V., Lange, M., Peidli, S. *et al.* Generalizing RNA velocity to transient cell states through dynamical modeling. *Nat Biotechnol* **38**, 1408–1414 (2020). <https://doi.org/10.1038/s41587-020-0591-3>

Decision Letter, first revision:

Our ref: NMETH-A52835A

13th Oct 2023

Dear Dr. Ren,

Thank you for submitting your revised manuscript "SnapATAC2: a fast, scalable and versatile tool for analysis of single-cell omics data" (NMETH-A52835A). It has now been seen by the original referees and their comments are below. The reviewers find that the paper has improved in revision, and therefore we'll be happy in principle to publish it in Nature Methods, pending minor revisions to satisfy the referees' final requests and to comply with our editorial and formatting guidelines.

TRANSPARENT PEER REVIEW

Nature Methods offers a transparent peer review option for new original research manuscripts

submitted from 17th February 2021. We encourage increased transparency in peer review by publishing the reviewer comments, author rebuttal letters and editorial decision letters if the authors agree. Such peer review material is made available as a supplementary peer review file. **Please state in the cover letter 'I wish to participate in transparent peer review' if you want to opt in, or 'I do not wish to participate in transparent peer review' if you don't.** Failure to state your preference will result in delays in accepting your manuscript for publication.

ORCID

Sincerely,
Lei

Lei Tang, Ph.D.
Senior Editor
Nature Methods

Reviewer #1 (Remarks to the Author):

My comments to the initial version of the manuscript have been satisfactorily addressed. I do not have more concerns to this revised and improved manuscript and recommend acceptance for publication.

Reviewer #2 (Remarks to the Author):

The authors have satisfactorily resolved all my comments. Congratulations on the great work, and the manuscript should be ready for publication.

Reviewer #3 (Remarks to the Author):

The authors have done thorough work in addressing previous concerns, and the revised manuscript has been substantially improved:

- providing a much better description of benchmarking.
- including new benchmark comparisons with other multimodal analysis methods.
- replacing most Umaps with quantitative assessments, making the value of the methods much clearer.

SnapATAC2 is a valuable software resource for single cell analysis, and is suitable for publication.

Author Rebuttal, first revision:

Reviewer #1:

Remarks to the Author:

My comments to the initial version of the manuscript have been satisfactorily addressed. I do not have more concerns to this revised and improved manuscript and recommend acceptance for publication.

We would like to thank the reviewers for their constructive comments.

Reviewer #2:

Remarks to the Author:

The authors have satisfactorily resolved all my comments. Congratulations on the great work, and the manuscript should be ready for publication.

We would like to thank the reviewers for their constructive comments.

Reviewer #3:

Remarks to the Author:

The authors have done thorough work in addressing previous concerns, and the revised manuscript has been substantially improved:

- providing a much better description of benchmarking.
- including new benchmark comparisons with other multimodal analysis methods.
- replacing most Umaps with quantitative assessments, making the value of the methods much clearer.

SnapATAC2 is a valuable software resource for single cell analysis, and is suitable for publication.

We would like to thank the reviewers for their constructive comments.

Final Decision Letter:

Dear Dr Ren,

I am pleased to inform you that your Article, "A fast, scalable and versatile tool for analysis of single-cell omics data", has now been accepted for publication in Nature Methods. Your paper is tentatively scheduled for publication in our Jan 2024 print issue, and will be published online prior to that. The received and accepted dates will be 9th Jun 2023 and 23rd Nov 2023. This note is intended to let you know what to expect from us over the next month or so, and to let you know where to address any further questions.

Over the next few weeks, your paper will be copyedited to ensure that it conforms to Nature Methods style. Once your paper is typeset, you will receive an email with a link to choose the appropriate publishing options for your paper and our Author Services team will be in touch regarding any additional information that may be required.

You will receive a link to your electronic proof via email with a request to make any corrections within

48 hours. If, when you receive your proof, you cannot meet this deadline, please inform us at rjsproduction@springernature.com immediately.

Please note that *Nature Methods* is a Transformative Journal (TJ). Authors may publish their research with us through the traditional subscription access route or make their paper immediately open access through payment of an article-processing charge (APC). Authors will not be required to make a final decision about access to their article until it has been accepted. [Find out more about Transformative Journals](https://www.springernature.com/gp/open-research/transformative-journals)

Your paper will now be copyedited to ensure that it conforms to Nature Methods style. Once proofs are generated, they will be sent to you electronically and you will be asked to send a corrected version within 24 hours. It is extremely important that you let us know now whether you will be difficult to contact over the next month. If this is the case, we ask that you send us the contact information (email, phone and fax) of someone who will be able to check the proofs and deal with any last-minute problems.

If, when you receive your proof, you cannot meet the deadline, please inform us at rjsproduction@springernature.com immediately.

Once your manuscript is typeset and you have completed the appropriate grant of rights, you will receive a link to your electronic proof via email with a request to make any corrections within 48 hours. If, when you receive your proof, you cannot meet this deadline, please inform us at rjsproduction@springernature.com immediately.

Once your paper has been scheduled for online publication, the Nature press office will be in touch to confirm the details.

Once your paper has been scheduled for online publication, the Nature press office will be in touch to confirm the details.

Content is published online weekly on Mondays and Thursdays, and the embargo is set at 16:00 London time (GMT)/11:00 am US Eastern time (EST) on the day of publication. If you need to know the exact publication date or when the news embargo will be lifted, please contact our press office after you have submitted your proof corrections. Now is the time to inform your Public Relations or Press Office about your paper, as they might be interested in promoting its publication. This will allow them time to prepare an accurate and satisfactory press release. Include your manuscript tracking number NMETH-A52835B and the name of the journal, which they will need when they contact our office.

About one week before your paper is published online, we shall be distributing a press release to news organizations worldwide, which may include details of your work. We are happy for your institution or funding agency to prepare its own press release, but it must mention the embargo date and Nature Methods. Our Press Office will contact you closer to the time of publication, but if you or your Press Office have any inquiries in the meantime, please contact press@nature.com.

Nature Portfolio journals [encourage authors to share their step-by-step experimental protocols](https://www.nature.com/nature-research/editorial-policies/reporting-standards#protocols) on a protocol sharing platform of their choice. Nature Portfolio 's Protocol Exchange is a free-to-use and open resource for protocols; protocols deposited in Protocol Exchange are citable and can be linked from the published article. More details can found at www.nature.com/protocolexchange/about.

Best regards,
Lei

Lei Tang, Ph.D.

Senior Editor
Nature Methods